# Pre-initiation and elongation structures of full-length La Crosse virus polymerase reveal functionally important conformational changes

Benoît Arragain 1, Grégory Effantin[1], Piotr Gerlach 2,3, Juan Reguera[2,4], Guy Schoehn 1, Stephen Cusack 2✉ & Hélène Malet 1✉

*Bunyavirales* is an order of segmented negative-strand RNA viruses comprising several life-threatening pathogens against which no effective treatment is currently available. Replication and transcription of the RNA genome constitute essential processes performed by the virally encoded multi-domain RNA-dependent RNA polymerase. Here, we describe the complete high-resolution cryo-EM structure of La Crosse virus polymerase. It reveals the presence of key protruding C-terminal domains, notably the cap-binding domain, which undergoes large movements related to its role in transcription initiation, and a zinc-binding domain that displays a fold not previously observed. We capture the polymerase structure at pre-initiation and elongation states, uncovering the coordinated movement of the priming loop, mid-thumb ring linker and lid domain required for the establishment of a ten-base-pair template-product RNA duplex before strand separation into respective exit tunnels. These structural details and the observed dynamics of key functional elements will be instrumental for structure-based development of polymerase inhibitors.

[1] Université Grenoble Alpes, CNRS, CEA, Institute for Structural Biology (IBS), F-38000 Grenoble, France. [2] European Molecular Biology Laboratory, Grenoble, France. [3] Present address: Department of Structural Cell Biology, Max Planck Institute of Biochemistry, Munich, Germany. [4] Present address: Aix-Marseille Université, CNRS, INSERM, AFMB UMR 7257, 13288 Marseille, France. ✉email: cusack@embl.fr; helene.malet@ibs.fr

**B**unyavirales is a very large and diverse order of segmented negative-strand RNA viruses (sNSV) comprising more than 500 species classified into 12 families[1]. It contains serious human pathogens such as La Crosse virus (LACV, *Peribunyaviridae* family), Hantaan virus (HTNV, *Hantaviridae* family), Crimean-Congo hemorrhagic fever virus (CCHFV, *Nairoviridae* family), Rift Valley Fever virus (RVFV, *Phenuiviridae* family), and Lassa fever virus (LASV, *Arenaviridae* family). Viruses from the *Bunyavirales* order are related to other sNSV and in particular to influenza virus, a major human pathogen belonging to the *Orthomyxoviridae* family.

Replication and transcription of sNSV viral genomic segments are performed by the virally encoded RNA-dependent RNA polymerase, also called L protein for *Bunyavirales*[2]. These processes are performed in the cytoplasm of infected cells for Bunyaviruses, whereas they occur in the nucleus for influenza virus[3,4]. Replication generates full-length genome or antigenome copies (vRNA and cRNA, respectively), whereas transcription produces capped viral mRNA that are recognized by the cellular translation machinery to produce viral proteins. Transcription is initiated by a "cap-snatching" mechanism, whereby host 5′ capped RNAs are bound by the L cap-binding domain (CBD), cleaved by the L endonuclease domain several nucleotides downstream, and then used to prime synthesis of mRNA[2,5,6].

Although the overall mechanism of transcription initiation is likely conserved between sNSVs, several elements suggest some divergence between viral families. First, the source of capped RNA differs. Whereas influenza polymerase interacts directly with the host RNA polymerase II to snatch the caps of nascent transcripts in the nucleus[7], bunyavirus polymerases act in the cytoplasm. It is currently unclear which cytoplasmic capped RNAs are accessed by bunyavirus polymerases, and in what context, and if the polymerase contains specific domains that interact with host capped-RNA-bound proteins. Second, the length of the host-derived capped RNA primer generated after cleavage by the endonuclease differs between families[5], 0–7 in *Arenaviridae*, 10–18 in *Peribunyaviridae*, 10–14 in *Orthomyxoviridae*, suggesting differences in the relative position of the endonuclease, CBD and polymerase active site. Third, CBD localization within L proteins remains unclear for several viral families primarily due to the absence of a definitive motif for the cap-binding site and because of the high divergence in sequence between polymerases, particularly in their C-terminal region. Identification of the CBD in the C-terminal region of L proteins has however recently been achieved for both Rift Valley Fever virus (RVFV, *Phenuiviridae*) and California Academy of Science virus (CASV, *Arenaviridae*), thanks to the determination of isolated CBD domain structures[8,9].

To understand the detailed mechanisms of replication and transcription, structures of the full-length polymerase are essential. Significant advances have recently been made on influenza polymerase with structures stalled at different steps of transcription now being available. They reveal that influenza CBD undergoes a 70° movement to bring the capped RNA from an orientation in which it can be cleaved by the endonuclease to one where it can enter the polymerase active site[10–12]. A snapshot of transcription elongation has been captured revealing the presence of a nine-base-pair template-product RNA inside the active site cavity that is then separated into two single-stranded RNAs exiting through separated tunnels[12]. In comparison, structural information on *Bunyavirales* polymerase remains limited with only the structure of a C-terminally truncated construct of LACV polymerase (residues 1–1750, LACV-L$_{1-1750}$) being currently available[13]. LACV-L$_{1-1750}$ is composed of an N-terminal protruding endonuclease domain (residues 1–185) and a polymerase core containing the RNA synthesis active site (residues 186–1750). It was solved by X-ray crystallography in the pre-initiation state in complex with the 5′ and 3′ promoter ends. Both promoter ends bind sequence-specifically in separate pockets away from the active site, respectively, called the "5′ end stem-loop pocket" and the "3′ end pre-initiation pocket". The LACV-L$_{1-1750}$ structure also depicts the presence of an active site cavity with typical polymerase motifs as well as distinct template and product exit tunnels.

To reveal the structure of the C-terminal region of LACV-L and the overall architecture of the complete polymerase, we determined the structure of full length LACV-L (LACV-L FL) by X-ray crystallography and high-resolution cryo-EM. We uncover the structure of LACV-L CBD, which contains a specific insertion allowing interaction with the endonuclease. We find that the extreme C-terminal region of LACV-L FL is a zinc-binding domain (ZBD) that is absent in other sNSV polymerases of known structure and may correspond to a host–protein interaction platform. We also capture snapshots of LACV polymerase in both pre-initiation and elongation-mimicking states, thereby revealing, amongst other conformational changes, the movement of the priming loop that unblocks the active site cavity and permits accommodation of a 10-base-pair template-product duplex, characteristic of elongation.

## Results

**Structure determination of LACV-L FL protein.** LACV-L FL was expressed in insect cells and purified to homogeneity based on the protocol described in Gerlach et al.[13] (Supplementary Fig. 1a). Slight modifications were however necessary in order to stabilize LACV-L FL, in particular the addition of nucleotides 1–16 of the 3′ vRNA (3′OH-UCAUCACAUGAUGGUU) and complementary 8-mer corresponding to the nucleotides 9–16 of the 5′vRNA (5′OH-GCUACCAA) prior to the decrease to 150 mM of NaCl concentration in the buffer. LACV-L was then further stabilized by the addition of the first 10 nucleotides of the 5′ vRNA (5′pAGUAGUGUGC), that were added by crystal soaking or just before cryo-EM grid freezing. LACV-L FL was crystallized and its structure solved at 4.0 Å resolution by molecular replacement using LACV-L$_{1-1750}$ as a template, revealing two molecules in the asymmetric unit. There was clear extra density showing repositioning of the endonuclease and for the previously missing C-terminal region, but the resolution was insufficient for building an accurate model (Supplementary Fig. 1b, Supplementary Table 1). LACV-L FL was subsequently characterized by cryo-EM, resulting in a 3.0 Å resolution structure. A 2.28 million particle dataset was collected on a Titan Krios equipped with a K2 direct electron detector (Supplementary Fig. 2a). 2D and 3D classifications revealed that the C-terminal region of the polymerase is extremely flexible and only 0.37 million particles displaying a defined density for the C-terminal region were kept for further structural analysis (Supplementary Fig. 2b). The resulting "stable dataset" was further 3D classified resulting in the separation of two defined states: (i) the expected pre-initiation state and (ii) an elongation-mimicking state in which the complementary 3′ and 5′ vRNA formed a double-stranded RNA that could be accommodated within the active site cavity (Fig. 1b and c, Supplementary Fig. 2c). Even though only the "stable dataset" was used for 3D classification, the C-terminal region (residues 1752–2263) remained poorly defined due to flexibility. Advanced image analysis (see "Methods" section) was necessary in order to determine the structure of all C-terminal domains between 3.0 and 3.5 Å resolution (Supplementary Figs. 2c, 3 and Supplementary Table 2). The complete model of LACV-L FL was manually built and refined (Fig. 1).

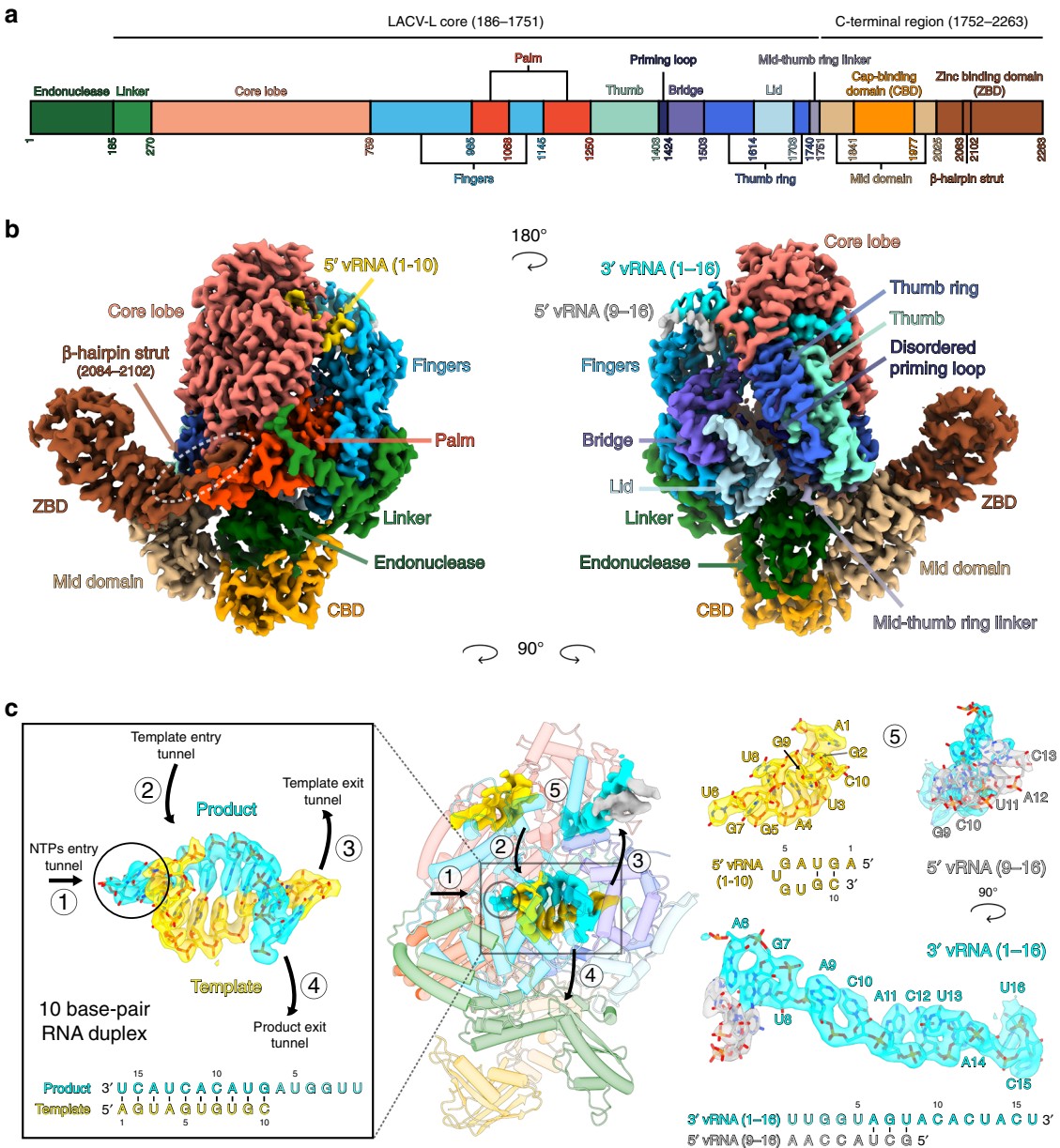

**Fig. 1 Cryo-EM structures of LACV-L FL. a** Schematic representation of LACV-L FL domain structure. **b** Two views of LACV-L FL cryo-EM map at pre-initiation stage. A composite cryo-EM map was assembled from individual maps of the (i) LACV-L core, endonuclease and mid domain, (ii) mid domain and CBD, and (iii) mid domain and ZBD. Domains and RNAs are indicated with arrows and colored as in **a**. **c** Cartoon representation of semi-transparent LACV-L FL cryo-EM structure at elongation-mimicking stage rotated of 90° compared to **b**. Close-up views of the Coulomb-potentials and models of all RNAs visible in the elongation-mimicking map. The position of the four tunnels are shown and numbered from 1 to 4. The 5′ end vRNA (1–10) in its "5′ end stem-loop loop pocket" is displayed in yellow, the 3′ vRNA (1–16) in its "3′ end pre-initiation pocket" is shown in cyan, the 5′ vRNA (9–16) that hybridizes with the 3′ vRNA (1–16) is colored in light gray. The RNA that mimics the template and product are shown in yellow and cyan, respectively. The sequence and secondary structures of nucleic acid moieties in each complex are displayed.

**Overall structure of LACV-L FL.** The X-ray and cryo-EM structures reveal the same overall arrangement of LACV-L FL. The structure of the polymerase core (residues 186–1751) is conserved compared to the LACV-L$_{1-1750}$ construct (RMSD of 0.474 Å on 1187 Cα) but the endonuclease domain undergoes a large rotational movement of 180° (Supplementary Fig. 4). The previously unobserved C-terminal region (1752–2263) protrudes away from the core and forms an elongated arc-shaped structure that includes the mid domain (residues 1752–1841 and 1978–2025), the CBD (residues 1842–1977), and the ZBD (residues 2026–2263) (Fig. 1a and b). The C-terminal region is supported and stabilized by a β-hairpin strut (residues 2084–2102)

that emerges from the ZBD domain and bridges the entire C-terminal region to the core (Fig. 1b).

**Endonuclease interactions with the other polymerase domains.** The endonuclease is held in place by hydrophobic interactions with a large number of residues from different domains (Fig. 2a). The N-terminus of the endonuclease and its un-cleaved TEV cleavage site are buried between the thumb ring (residues 1716, 1717, 1720) and the residue 714 of the core lobe β-hairpin (Fig. 2b and a). The α-helices 6 and 7 of the endonuclease face the palm domain (α-helix 38) and the finger domain (α-helix 30) (Fig. 2a

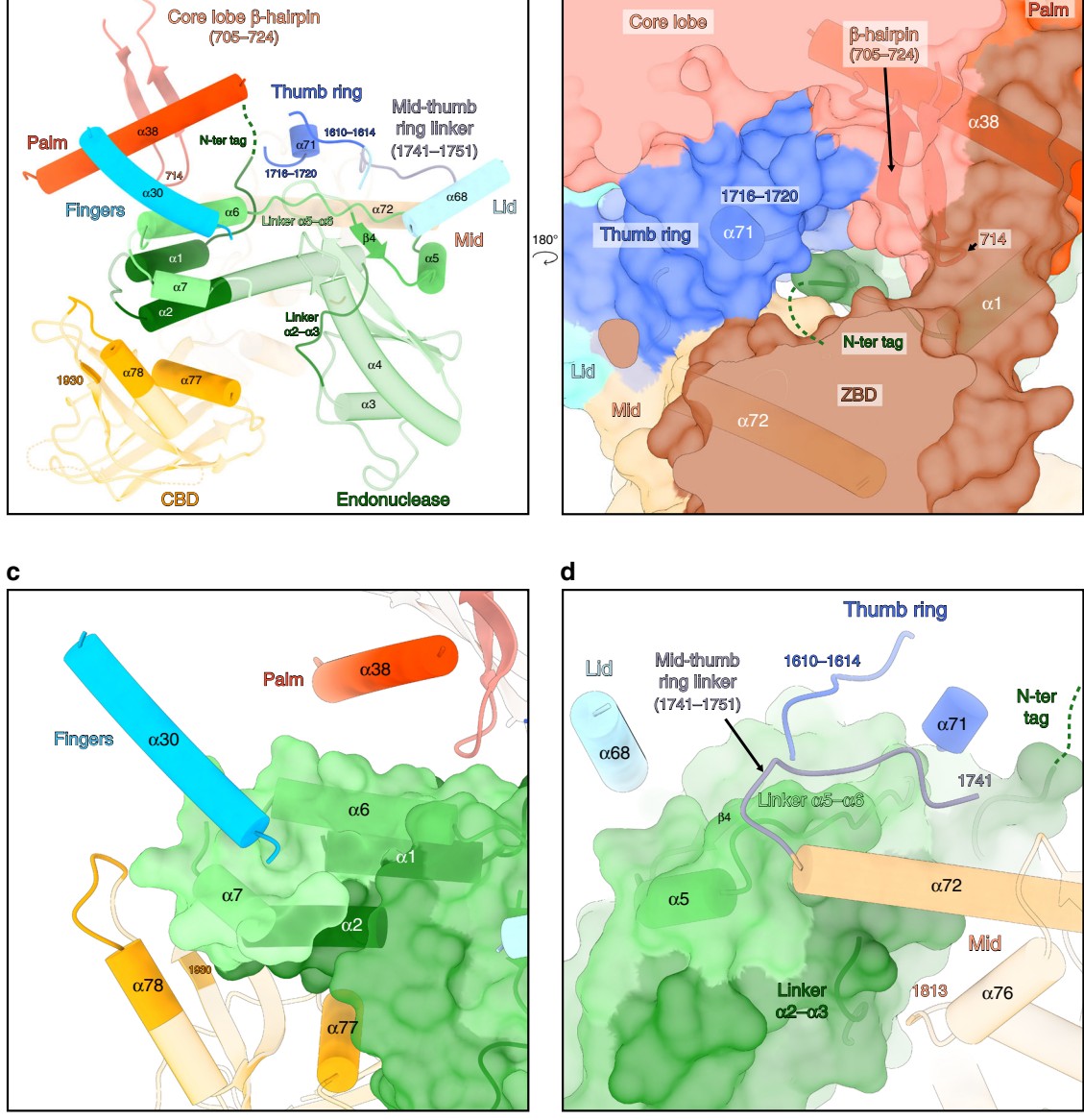

**Fig. 2 Interactions of the endonuclease with the other polymerase domains. a** Overview of the endonuclease interactions with the other polymerase domains. LACV-L orientation is the same as in Fig. 1c. Secondary structures and residues interacting with different parts of the endonuclease are shown in non-transparent and labeled. The endonuclease is colored from dark green on the N-terminus to light green on the C-terminus. The core lobe, fingers, thumb ring, lid, mid-thumb ring linker, mid, CBD are colored as in Fig. 1a. **b** Close-up view of the endonuclease N-terminus. Transparent surface of the polymerase is shown. The secondary structures of the core lobe and thumb ring domains that interact with the N-terminal tag are displayed, labeled and colored as in **a**. The N-terminal tag is represented as a dark green dotted line. **c** Close-up view of the interactions between the endonuclease and the following LACV-L domains: fingers, palm and CBD. The endonuclease is displayed as a transparent surface and colored as in **a**. **d** Close-up view of the interactions between the endonuclease and the following LACV-L domains: thumb ring, lid, mid-thumb ring linker and mid. The endonuclease is displayed as in **c**. The secondary structures are labeled and the N-terminal tag is represented as a dark green dotted line.

and c). Hydrophobic interactions are depicted between residues 46–53 (corresponding to the linker between α-helices 2 and 3) and the mid domain (residue 1813 and residues of the α-helix 76) (Fig. 2d). Residues 137–159 of the endonuclease (corresponding to the β-strand 4, the α-helix 5 and the linker between the α-helices 5 and 6) interact with the residues 1610–1614 of the thumb ring domain, the α-helix 68 of the lid domain, the α-helix 71 of the thumb ring domain, the mid-thumb ring linker (residues 1741–1751), and the α-helix 72 of the mid domain (Fig. 2d). Finally, the endonuclease interacts with the CBD (Fig. 2a), through interactions that change depending on the CBD position (described in the paragraph below).

**Structure and mobility of the cap-binding domain of LACV-L.** The CBD is composed of a five-stranded anti-parallel β-sheet (β34, β35, β36, β37, β41) packed against the α-helix 77 that is flanked by a three-stranded antiparallel β-sheet (β38, β39, β40), the α-helix 78 and long loops (Fig. 3a). There is a disordered loop between the first two strands of the CBD five-stranded β-sheet (Fig. 3 a, b) that contains a number of residues highly conserved in all *Peribunyaviridae* L proteins, although there is no density for them. The $m^7$GTP-binding sites of RVFV and influenza virus CBD[8,14] which share the same overall fold are located in an equivalent loop (Supplementary Fig. 5). The $m^7$GTP cap-binding site of LACV-L can therefore be predicted to be composed of

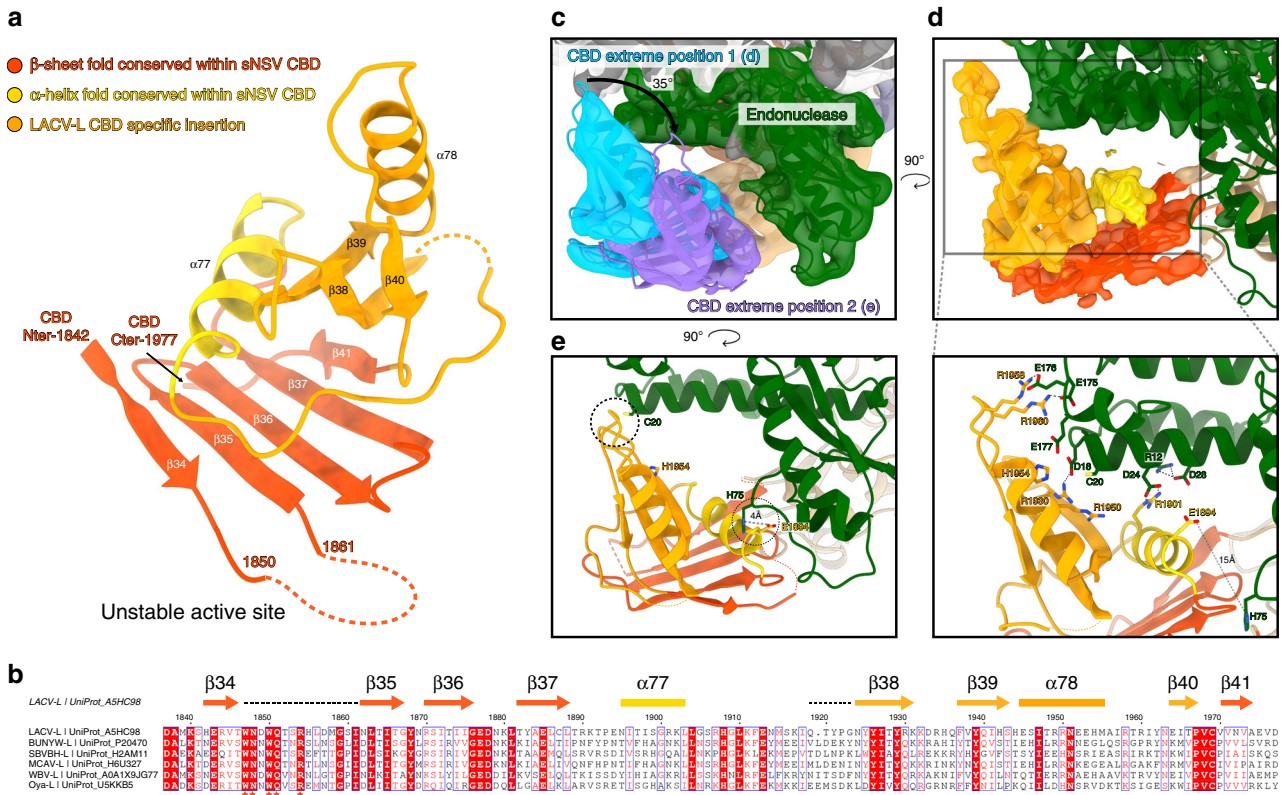

**Fig. 3 LACV-L CBD structure and its interaction with the endonuclease. a** LACV-L CBD atomic model. Secondary structures are shown. The fold conserved with other sNSV CBD is shown in red and yellow. LACV CBD insertion is shown in orange. The missing loop comprising the active site is shown as a dotted line. **b** Multiple alignment of six *Peribunyaviridae* CBD: La Crosse virus (LACV), Bunyamwera virus (BUNYW), Schmallenberg (SVBVH), Macaua virus (MCAV), Wolkberg virus (WBV), and Oya virus. Secondary structures of LACV-L CBD are shown and colored as in **a**. Missing residues of LACV-L CBD are presented as dotted lines. Conserved residues of *Peribunyaviridae* CBD active site motif (WNxWQxxR) are shown as orange stars. **c** Cryo-EM 3D classes corresponding to LACV-L CBD extreme position 1 and 2 are superimposed. CBD movement is compared to LACV-L core and endonuclease that adopt stable positions. Their CBD are, respectively, shown in blue and purple. The endonuclease domain is shown in green. LACV-L core is shown in gray. **d** Overview and close-up view of the CBD/endonuclease domain interactions in the extreme position 1. Interacting residues are identified and labeled. CBD coloring is the same as in **a**. **e** Overview of the CBD/endonuclease domain interaction in the extreme position 2. Interacting residues are identified and labeled. CBD side chain positions remain however hypothetical due to the CBD EM map resolution in extreme position 2.

W1847 and/or W1850 that would stack the guanine moiety of the m⁷GTP. This interaction would be supported by Q1851 and R1854 that would, respectively, interact with the guanine and the phosphates (Fig. 3b, Supplementary Fig. 5). This suggests a conserved mode of m⁷GTP interaction mediated by functionally equivalent residues without any overall sequence conservation for LACV, RVFV, and influenza CBDs.

The CBD is rotationally mobile as visualized in a 3D variability analysis of the dataset (Supplementary Movie 1). Its large movement is enabled by the conformationally stable mid domain that acts as a central hub mediating contacts between the core, the CBD, and the ZBD (Fig. 4a). Several CBD positions can be separated by 3D classification (Supplementary Fig. 2c, see "Methods" section) and a rotation of 35° is visible between the two extreme positions (Fig. 3c). In the extreme position 1, residues 12–28 and 175–178 of the endonuclease domain interact with residues E1894, R1901, R1930, 1950–1960 of the CBD mainly through electrostatic interactions (Fig. 3d). In the extreme position 2, the contacts between the CBD and the rest of the polymerase are rather sparse, explaining its instability (Fig. 3e). The only interactions are mediated by the loop 1932–1936 of the CBD that is proximal to C20 of the endonuclease domain, and the residue E1894 of the CBD that is close to the H75 of the endonuclease domain (Fig. 3e).

**Structural organization of the zinc-binding domain.** The C-terminal extremity of LACV-L is an α-helical domain with a long protruding β-hairpin (Fig. 4). Its two equivalently sized sub-domains surround a metal ion that is coordinated by the residues C2064, H2169, D2178, and H2182, suggesting that it is a zinc ion (Fig. 4a). These four residues occur unchanged in all the 84 *Peribunyaviridae* sequences deposited in the NCBI, indicating that ion binding is a conserved feature in this viral family. Whereas zinc ion binding by viral polymerases is rather common[15–17], the overall topology of the LACV ZBD has not been previously observed according to a DALI search[18]. LACV ZBD protrudes away from the polymerase, suggesting that it could be extremely mobile. This appears to be the case for many of the particles, impeding their use in structure determination of this domain (Supplementary Fig. 2c). However, in the particles used for high-resolution cryo-EM determination of the C-terminal region and in the X-ray structure, the above-mentioned protruding β-hairpin strut (residues 2084–2102, Fig. 4a) stabilizes the orientation of the ZBD with respect to the core by forming a four-stranded antiparallel β-sheet together with a β-hairpin from the core lobe (residues 705–724) (Fig. 4b). Interestingly, the β-hairpin was not visible in the LACV-L₁₋₁₇₅₀ electron density and is only structured in the presence of the ZBD β-hairpin. In addition, the ZBD β-hairpin strut makes several hydrophobic interactions with

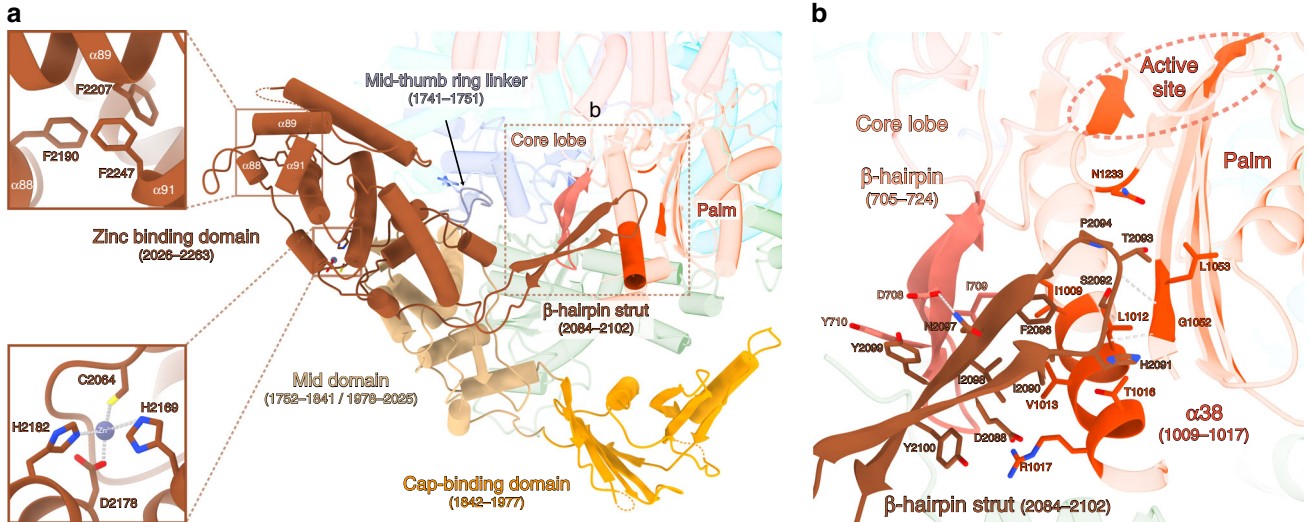

**Fig. 4 C-terminal region of LACV-L FL. a** The structure of LACV-L C-terminal region is shown as non-transparent, the rest of the polymerase is shown as transparent, except the residues interacting with the ZBD β-hairpin strut. The domains are colored as in Fig. 1. Position of the last α-helix 91 that interacts with α-helices 88/89 is indicated and a close-up view of hydrophobic amino acids interaction is shown. Close-up view on the coordination of the presumed zinc ion from the ZBD. Conserved residues implicated in ion coordination are shown and labeled. The ZBD β-hairpin strut that protrudes towards the core is surrounded by a dotted rectangle corresponding to panel **b** view. **b** Close-up view of the ZBD β-hairpin strut. Residues from the core lobe β-hairpin (705–724) and the palm domain that interact with the ZBD β-hairpin are indicated. The active site is surrounded by a dotted line.

residues 1009–1017, L1053, and N1233 of the palm domain, proximal to the polymerase active site (Fig. 4b). The extreme C-terminal α-helix 91 of the ZBD (Fig. 4a and Supplementary Fig. 1c) is connected to the rest of the domain via a long flexible loop permitting large movements. In the crystal structure, α-helix 91 protrudes away to bind to a hydrophobic pocket present in the ZBD of the second polymerase of the asymmetric unit, forming a domain-swap dimer (Supplementary Fig. 1c). In the cryo-EM map, the polymerase is monomeric and some density present at low threshold suggests that the α-helix 91 may fold back into the same hydrophobic pocket of the ZBD, although the binding might be rather labile (Fig. 4a).

**Elongation-mimicking state**. Based on the RNA promoter sequences with which the polymerase was incubated, we expected to obtain only the pre-initiation state. However, extensive 3D classification resulted in identification of an alternative RNA-bound subset of particles in which a 10-base-pair duplex is unexpectedly visible in the active site cavity (Supplementary Fig. 2c). This structure, mimicking an elongation state with a bound product–template duplex, is determined at 3.0 Å resolution, enabling to distinguish unambiguously purine and pyrimidine bases (Fig. 5a). It can thus be deduced that the RNA duplex corresponds to the hybridization between the nearly complementary 5′ and 3′ promoter ends (5′p-(1)AGUAGU-GUGC(10) and 3′OH-(16)UCAUCACAUG(7)), corresponding to nucleotides 1–10 for the 5′ and 7–16 for the 3′ (Fig. 5a, c, d). Visualization of this state shows that a small fraction of the "stable dataset subset" (59,152 particles out of 370,497, Supplementary Fig. 2c) was able, in the in vitro conditions used and with 3′ and 5′ promoter ends in excess, to internalize the promoter duplex in the active site cavity. This is in addition to the 3′ and 5′ RNA promoter ends also being bound in their respective "3′ end pre-initiation pocket" and "5′ stem-loop pocket", in positions identical to the ones observed at pre-initiation, showing the RNA binding compatibility between all these separate RNA binding sites (Fig. 1c). Although not a true elongation state, the structure obtained fortuitously mimics this state and gives insight into the mechanisms of (i) RNA binding in the active site

cavity and (ii) template–product separation after formation of a 10-base-pair double-stranded RNA in the active site cavity.

The backbone and some bases of the template–product duplex contact many residues lining the polymerase active site chamber via both van der Waals and polar interactions (Fig. 5c, d). The active-site proximal part of the template-mimicking RNA (nucleotides 1, 2 and 3) interacts with the finger domain, the central part (nucleotides 4, 5) binds to the palm, while the distal part (nucleotides 6–10) interacts with the thumb and the thumb ring domains (Fig. 5a, c, d). The proximal part of the product-mimicking RNA (nucleotides 14–16) is surrounded by the palm domain, the central part of the product (nucleotides 10–13) interacts with the core and the core lobe, while the distal part of the product-mimicking RNA (nucleotides 7–9) mainly binds to the bridge and the finger domains (Fig. 5a, c, d). The LACV-L catalytic core shares with other viral RNA-dependant RNA-polymerases the six conserved structural motifs (A–F)[19] (Fig. 5b). In addition, motifs G and H that are specific to sNSV polymerases are also visible[13] (Fig. 5b).

The polymerase conformation mimics a post-incorporation, pre-translocation elongation step in which an incoming nucleotide would just have been incorporated into the product. During the nucleotide addition cycle, viral polymerase active sites undergo small structural changes that enable NTP-binding, NTP-incorporation, and subsequent RNA translocation[20]. Whereas a particular configuration of (i) the nucleotide to be incorporated, (ii) the product RNA to be elongated, (iii) two magnesium ions, and (iv) the three aspartic acids of motifs A and C is necessary for catalysis of the phosphoryl transfer reaction, the organization of the active site changes subsequently[20]. Such post-incorporation structural changes are visible in LACV-L elongation mimicking stage. Perhaps related to the fact that there is no pyrophosphate (since there was no reaction), the two magnesium ions that control nucleotide addition are not present in the catalytic configuration. Instead, a presumed $Mg^{2+}$ ion, coordinated by residues D1188 (motif C), E1237 (motif E), and the carbonyl oxygen of A1059 (motif A), is present, typical of the inactive open state of the polymerase active site (Fig. 5b, Supplementary Fig. 6a). Some other motifs are also in post-incorporation conformation,

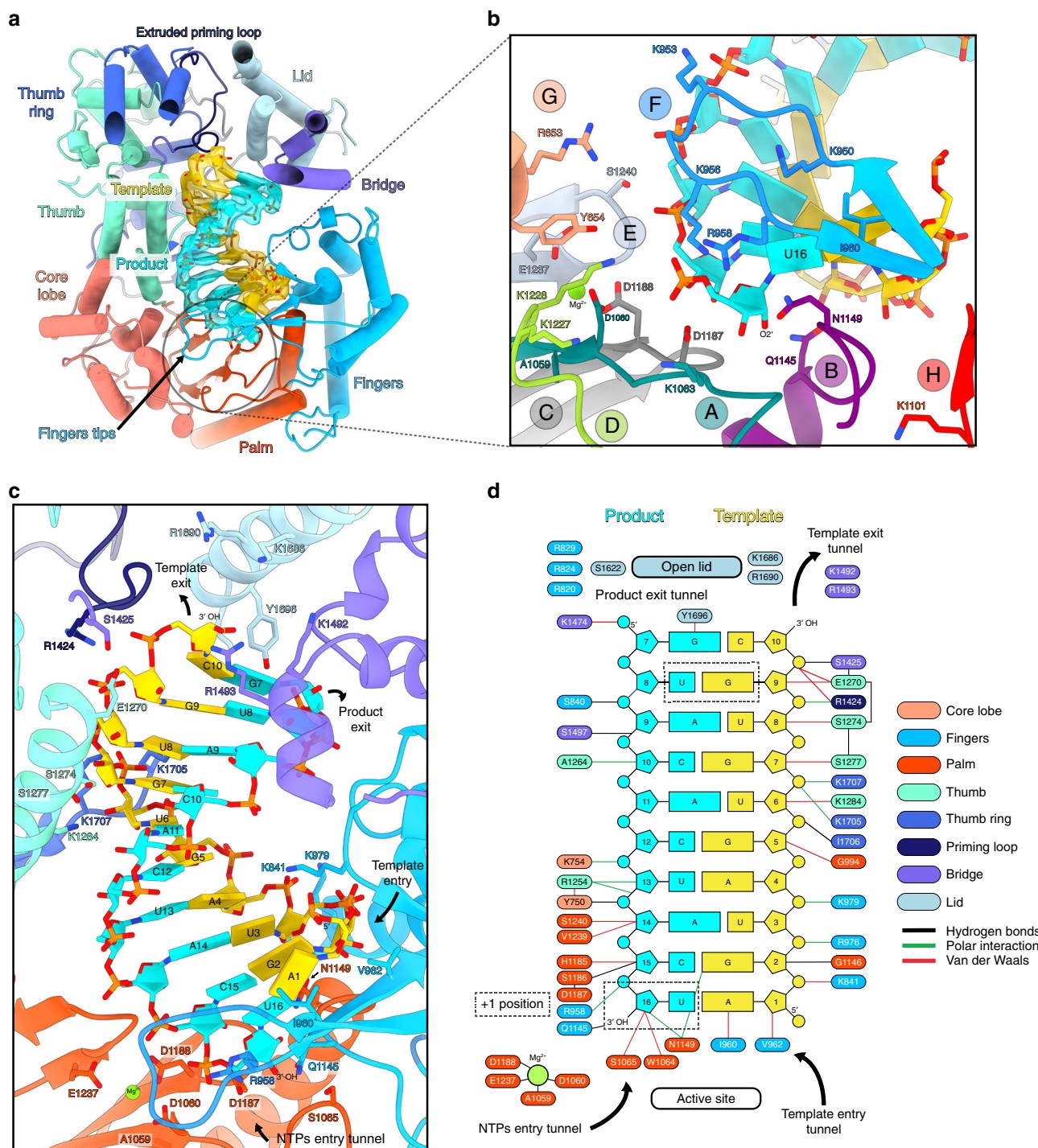

such as the fingertips residues R958 and I960 that, respectively, stack the bases of the product and template nucleotides in the +1 position, thereby stabilizing them (Fig. 5b). The motif B loop, which is implicated in the selection of the correct nucleotide to be incorporated, also adopts a conformation compatible with a post-incorporation state. Its residue Q1145 contacts the +1 position nucleotide base of the last incorporated product nucleotide, while residue N1149 interacts with the 2′ hydroxyl group of the template nucleotide (Fig. 5b). In summary, the elongation-mimicking structure represents a post-incorporation, pre-translocation elongation step containing a 10-base-pair template-product RNA in the active site chamber.

**Conformational changes between pre-initiation and elongation.** Comparison between the pre-initiation and elongation-mimicking states reveals key movements of the L protein in action. The priming loop (residues 1404–1424) is an essential element that usually stabilizes the first 'priming' nucleotide of the product during replication initiation. In the pre-initiation structure, this loop protrudes towards the active site but is disordered probably due to the absence of RNA and nucleotides (Fig. 6a). As part of the initiation to elongation transition, it extrudes from the active site via the template exit tunnel, thereby freeing space for the 10-base-pair RNA to fit in the active site chamber (Fig. 6b). The fully ordered and extruded priming loop is located on the

**Fig. 5 Cryo-EM structure of the LACV-L FL at elongation-mimicking stage. a** Cut-away view of the LACV-L FL at elongation-mimicking stage. Its orientation corresponds to a top view visualization of Fig. 1b left. The main domains are depicted and colored as in Fig. 1. The Coulomb potential map of the 10-base-pair product–template RNA is shown in cyan for the product (3′OH-UCAUCACAUG, nucleotides 7–16) and gold for the template (5′p-AGUAGUGUGC, nucleotides 1–10). The extruded priming loop is shown in dark blue. The active site position is indicated as a dotted circle. **b** View of the LACV-L FL active site showing the conserved RNA-dependent RNA polymerase functional motifs A–H (G and H are only conserved in sNSV polymerases). They are respectively colored turquoise, purple, gray, light green, light blue, blue, beige, and red for A–H. Template-mimicking and product-mimicking RNA are colored as in **a**. Presumed magnesium ion, is shown as a green sphere. **c** Interactions of the 10-base-pair product–template RNA in LACV-L active site cavity. Principal residues from the active site in palm domain (A1059, D1060, S1065, D1187, D1188, E1237), fingertips (R958), fingers (K841, I960, V962, K979, Q1145), priming loop (R1424), bridge (S1425, K1492, R1493), thumb (E1270, S1274, S1277, K1284), and lid (K1686, R1690, Y1696) are displayed. NTPs entry, template entry, template entry/exit tunnel directions are shown. Ion position is shown as in **b**. Nucleotides are labeled according to RNA promoter sequence. **d** Schematic representation of RNA–protein contacts in the active site cavity. Residues are colored according the domain to which they belong. Template and product RNA are numbered according to their position in the 5′ end promoter (5′p-AGUAGUGUGC, nucleotides 1–10) and the 3′ end promoter (3′OH-UCAUCACAUG, nucleotides 7–16). The U-G mismatch that is due to the non-perfect complementarity between the 5′ and 3′ promoters is surrounded by a dotted rectangle. Interaction type are color coded as indicated. Ion is shown as a green circle. Active site and lid domain positions are indicated. Nucleotide U16 corresponds to the nucleotide in position +1 of the product and is identified as such.

surface of the thumb ring and lid domains (Fig. 6b) with which it interacts mainly through hydrophobic contacts involving residues V1572, Y1576, A1751, M1753, N1658, L1661 (Fig. 6d). Interestingly, the priming loop movement is coupled with the reorganization of mid domain residues 1752–1761 from an α-helix to an extended loop. This results in an extension of the mid-thumb ring linker from residues 1741–1751 at pre-initiation to 1741–1761 at elongation. As a result of the helix unwinding, the mid-thumb ring linker loop extremity (residues 1750–1753, at pre-initiation), is displaced by 8 Å in the transition to elongation (Fig. 6d) and interacts with the priming loop residues 1414–1418 mainly through hydrophobic contacts (Fig. 6d).

The transition from pre-initiation to elongation is also coupled to coordinated domain movements. The lid domain rotates by 12° compared to the thumb ring, resulting in the opening of the template and the product exit tunnels (Fig. 6c). Separation of the template–product RNA duplex is made possible by the α-helix 70 of the lid domain that faces the distal part of the double-stranded RNA. Its residue Y1696 interacts with the product 5′ end nucleotide, thereby forcing strand-separation of the RNA duplex (Fig. 5c). Domain movements occurring between pre-initiation and elongation are nicely captured by a 3D variability analysis of the dataset (Supplementary Movie 2). It reveals a coordinated rotation of the endonuclease and the C-terminal region compared to the core, using the mid domain as a hinge, and resulting in 4.5 and 8 Å displacement of the endonuclease and the ZBD respectively (Supplementary Movie 2).

## Discussion

The structure presented here reveals the organization of the entire LACV-L protein. The newly described C-terminal domains can be compared with equivalent parts of *Phenuiviridae*, *Arenaviridae*, and *Orthomyxoviridae* polymerases. LACV-L CBD shares a conserved fold with equivalent structures from RVFV (*Phenuiviridae*)[8], CASV (*Arenaviridae*)[9], and influenza virus (*Orthomyxoviridae*)[14], consisting of an antiparallel β-sheet stacked against an α-helix (Supplementary Fig. 5). In addition, LACV-L CBD contains a family-specific insertion consisting of a three-stranded β-sheet (β38, β39, β40), α-helix 78 and charged loops (1932–1936 and 1956–1963). This insertion is likely related to the role of the CBD in transcription initiation as it is involved in interactions with the endonuclease (Fig. 3d). LACV C-terminal region also contains a mid domain and a ZBD. The mid domain fold is conserved between *Orthomyxoviridae* and *Arenaviridae* polymerases[9,11] (Supplementary Fig. 7b), whereas the distal C-terminal region differs between the three families. *Orthomyxoviridae* and *Arenaviridae*, respectively, have a PB2 627-domain and D1-III domain that are structurally related[9,11], whereas the

ZBD present in LACV has a different fold. This suggests that *Orthomyxoviridae* and *Arenaviridae* polymerases are more closely related (Supplementary Fig. 7c). The presence of the ZBD or PB2 627-domain in the C-terminal region in a protruding position compared to the polymerase core suggests that, as for the PB2 627-domain[21], LACV-L ZBD may play a role as a platform to recruit host cytoplasmic proteins. For instance it could interact with host proteins that bind capped RNA, host factors that mediate the transcription–translation coupling observed in *Bunyavirales*[5,22] or be involved in replication-related activities. It should however be noticed that, contrary to influenza PB2 627-domain, it is not possible to identify a non-conservative mutation of a residue in LACV-L ZBD that would differentiate one LACV strain to another and might reflect host specificity.

The endonuclease and the CBD have essential roles in transcription initiation. For influenza polymerase, structures depicting their coordinated movements explain how cap-dependent transcription is initiated[12,23]. The present LACV-L FL structure reveals two relative positions of the endonuclease and the CBD with respect to the core, that we call "LACV-L conformation 1" and "LACV-L conformation 2" (Supplementary Fig. 8a, Fig. 3c). In both conformations, the endonuclease is stabilized in position by making hydrophobic interactions with the thumb ring, core lobe, mid, palm, finger and cap-binding domains (Fig. 2). Its orientation however differs by around 180° compared to LACV-L$_{1-1750}$ (Supplementary Fig. 4) where the endonuclease protrudes away from the polymerase core, making only few interactions with it. Concerning the CBD, its position in the LACV-L FL dataset is variable, with extreme conformations being 35° apart, respectively, corresponding to LACV-L conformation 1 and LACV-L conformation 2 (Fig. 3c, Supplementary Fig. 8a, Supplementary Movie 1).

These conformations are likely to correspond to functional states. LACV-L conformation 1 may correspond to a pre-initiation transcription state compatible with the binding of cellular capped RNA, as its CBD is exposed towards the exterior (Supplementary Fig. 8a). In LACV-L conformation 2, the cap-binding site is closer to the endonuclease and we speculate that in this conformation the RNA can be cleaved by the endonuclease (Supplementary Fig. 8a). However, the observed location of the endonuclease prevents rotation of the CBD into a position where it can direct the capped primer into the active site for transcription initiation (Supplementary Fig. 8a), so a relative re-positioning of the two domains is expected. In addition, the endonuclease active site is located close to the product exit (Supplementary Fig. 8a). In order to prevent degradation of the product when it exits the polymerase, one can speculate that a large movement of the endonuclease may be necessary at a late elongation stage.

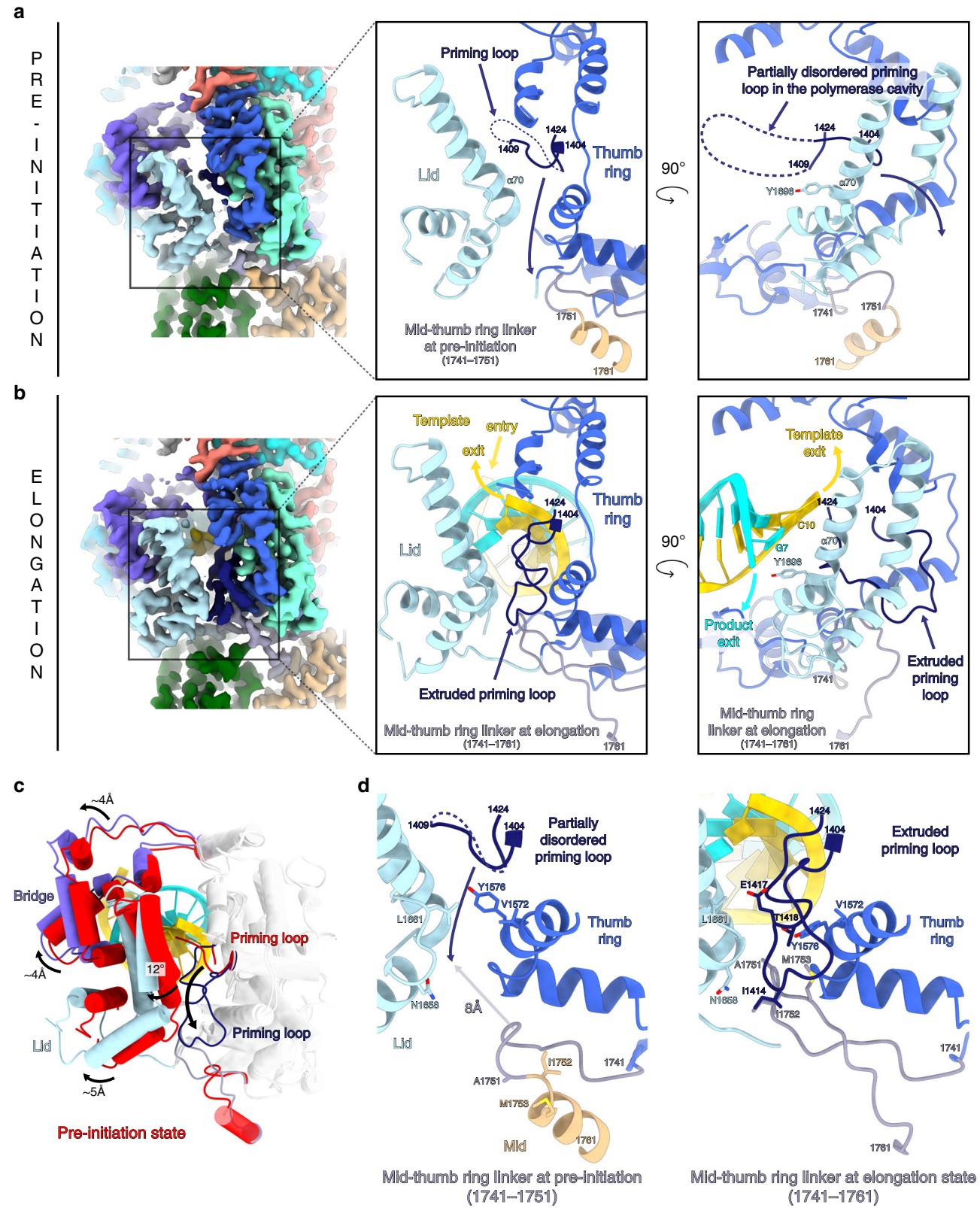

The position of both the endonuclease and the CBD in LACV-L conformation 1 and 2 significantly differ from the observed disposition of these domains in all the functional states described for influenza polymerase (Supplementary Fig. 8b). They also diverge from the recently described structures of full-length L protein from Machupo virus (*Arenaviridae* family)[17] and Severe Fever with Thrombocytopenia Syndrome virus L (*Phenuiviridae* family)[24,25] published while this manuscript was under review (Supplementary Fig. 8c, d). However, without being sure that these latter structures represent functionally active states, it is too

**Fig. 6 Conformational changes between pre-initiation and elongation-mimicking states. a** LACV-L FL in pre-initiation state. Left panel: cryo-EM map colored as in Fig. 1. Middle panel: close-up view. Right panel: close-up view rotated 90° compared to the middle panel. For more clarity, only the lid, thumb ring, and mid domains are shown in the middle and right panels and are colored in light blue, blue, and beige. The mid-thumb ring linker (1741–1751) is shown in purple gray. The priming loop is shown in dark blue. All these elements are labeled. The lid domain is closed and the residue Y1696 from the α-helix 70 points towards the cavity. The priming loop is disordered between residues 1410 and 1423 (dotted line). Priming loop movement occurring during elongation is depicted with a dark blue arrow. **b** LACV-L FL in elongation-mimicking state. Equivalent panels as in **a** are shown with domains and loops colored as in **a**. The mid-thumb ring linker contains residues 1741–1761 at elongation. The template tunnel entry and exit, the product exit are shown. The lid domain is open and Y1696 from the α-helix 70 interacts with the product-mimicking RNA. The priming loop is ordered and interacts with the lid, the thumb ring, and the mid-thumb ring linker. **c** Superimposition of the thumb and thumb ring (shown in transparent gray) in pre-initiation and elongation-mimicking state of LACV-L FL. The bridge, the lid, and the priming loop in pre-initiation are shown in red. The bridge, the lid and the priming loop in elongation-mimicking state are shown in purple, light blue and dark blue. Their rotation between the two states is labeled. Movement of the priming loop between the two states is shown with an arrow. **d** Priming loop and mid-thumb ring linker at pre-initiation (left) and elongation (right). Numbering of both elements are indicated. Residues from the lid and thumb ring that interact with the priming loop at pre-initiation and/or elongation are shown. The 8 Å displacement between the mid-thumb ring linker extremity at pre-initiation and elongation is indicated with an arrow.

early to draw any conclusions on differences in mechanism between these various viral polymerases. The insight provided here on LACV-L constitutes a basis to address in more detail the exact mechanisms underlying *Peribunyaviridae* transcription initiation in the future, notably by determining structures in complex with capped RNA.

The active sites of LACV and influenza polymerases in presence of RNA are remarkably similar (Supplementary Fig. 6). Previous studies of influenza polymerase have depicted the post-incorporation pre-translocation (Supplementary Fig. 6b)[23] and post-incorporation post-translocation (Supplementary Fig. 6c)[12] states. LACV-L FL bound to duplex RNA appears to mimic an intermediate between these two states as it displays a product nucleotide at the +1 position (as if it had been newly incorporated) with a magnesium ion and the catalytic aspartic acids in an inactive open state conformation (Supplementary Fig. 6a). Such an intermediate resembles the equivalent state in poliovirus polymerase (PDB: 3OL9)[26] although the observed movements of the active site between pre/post translocations are restricted to side chains movements in LACV/influenza polymerase structures, whereas they involve larger movements of motifs in poliovirus polymerase. It should however be noticed that the observed state in LACV-L was obtained by internalization of a double-stranded RNA and might not exactly reflect a true elongation intermediate notably because of the absence of pyrophosphate.

More generally, the cavities that bind the template-product RNA duplex are very similar in LACV-L and influenza polymerases. The mechanism of template–product separation by the lid domain is remarkably conserved with the equivalent helix being implicated[12]. The tyrosine that prevents double-strand continuation strand differs however slightly in position: Y1696 of LACV-L interacts with the RNA product whereas Y207 of influenza PB2 stacks with the last nucleotide of the template (Supplementary Fig. 9). Movements occurring between pre-initiation and elongation also differ between the two viral polymerases. For instance, the priming loop takes a completely different position and organization when it extrudes from the active site in the two cases (Supplementary Fig. 9). Influenza polymerase priming loop, which is 35-amino acid long, has 17 residues extruded into the solvent in a disordered loop, and projects towards the PB1–PB2 interface helical bundle[12] (Supplementary Fig. 9b). LACV-L priming loop, which is 20-amino acid long, orders itself at the surface of thumb ring and lid domains (Supplementary Fig. 9a). Its extrusion is coupled with a *Peribunyaviridae*-specific unwinding of the α-helix 72 of the mid-domain into a loop, enabling an interaction between the priming loop and the extended mid-thumb ring linker (Fig. 6d). Conformational changes of the whole C-terminal region upon the transition from pre-initiation to elongation visualized in the 3D variability

analysis (Supplementary Movie 2) have also not been observed in influenza polymerase.

Altogether, the complete structure of LACV-L FL captured in pre-initiation and elongation-mimicking states and the rotational mobility of the CBD provide mechanistic insight into bunyaviral transcription. This, reinforced by the atomic details of the polymerase active site, establishes a firm basis for future structure-based drug design that could target essential activities or critical conformation changes of *Peribunyaviridae* L proteins.

## Methods

**Cloning, expression, and purification**. Sequence-optimized synthetic DNA encoding a N-terminal his-tag, a TEV protease recognition site, and the LACV-L FL (strain LACV/mosquito/1978, GenBank: EF485038.1, UniProt: A5HC98) was synthetized (Geneart) and cloned into a pFastBac1 vector between NdeI and NotI restriction sites (Supplementary Table 3). The LACV-LFL expressing baculovirus was generated via the standard Bac-to-Bac method (Invitrogen). For large-scale expression, *Trichoplusia ni* High 5 cells at $0.5 \times 10^6$ cells/mL concentration were infected by adding 0.1% of virus. Expression was stopped 72 h after the day of proliferation arrest. The cells were disrupted by sonication for 3 min (10 s ON, 20 s OFF, 50% amplitude) on ice in lysis buffer (50 mM Tris–HCl pH 8, 500 mM NaCl, 20 mM Imidazole, 0.5 mM TCEP, 10% glycerol) with EDTA-free protease inhibitor complex. After lysate centrifugation at $48,384 \times g$ during 45 min at 4 °C, protein from the soluble faction was precipitated using $(NH_4)_2SO_4$ at 0.5 mg/ml and centrifuged at $104,630 \times g$ for 45 min, 4 °C. Supernatant was discarded, proteins were resuspended back in the same volume of lysis buffer and centrifuged at $104,630 \times g$ during 45 min at 4 °C. LACV-L FL was purified from the supernatant by nickel ion affinity chromatography after a wash step using 50 mM Tris–HCl pH 8, 1 M NaCl, 20 mM Imidazole, 0.5 mM TCEP, 10% glycerol and eluted using initial lysis buffer supplemented by 300 mM Imidazole. LACV-L FL fractions were pooled and dialyzed 1 h at 4 °C in heparin-loading buffer (50 mM Tris–HCl pH 8, 250 mM NaCl, 0.5 mM TCEP, 10% glycerol). Proteins were loaded on heparin column and eluted using 50 mM Tris–HCl pH 8, 1 M NaCl, 0.5 mM TCEP, 5% glycerol. LACV-L FL was then mixed in a 1:3 molar ratio with both 3′ (1–16, 3′-UCAUCACAUGAUGGUU-5′) and 5′ (9–16, 5′ GCUACCAA-3′) vRNA oligonucleotide ends which had been pre-annealed by heating at 95 °C for 2–5 min followed by cooling down on bench at room temperature. During overnight dialysis at 4 °C in a gel filtration buffer (20 mM Tris–HCl pH 8, 150 mM NaCl, 2 mM TCEP) LACV-L FL formed a complex with vRNA, which was ultimately resolved on the S200 size exclusion chromatography column.

**Crystallization and X-ray crystallography**. For crystallization, LACV-L FL in complex with pre-annealed 3′ (1–16) and 5′ (9–16) vRNA was concentrated to 5 mg/ml. The 5′ (1–10) vRNA end (5′-AGUAGUGUGC-3′) was later soaked into crystals in 1:2 molar ratio. Initial hits were dense and round precipitates that appeared in 100 mM Tris pH 8.0, 100 mM NaCl, and 8% PEG 4000. Upon manual reproduction in hanging drops, they grew as thin hexagonal plates, but were soft and fragile and diffracted only to ~8 Å. To improve the resolution, crystals were soaked in a stepwise manner with increasing concentration of the glycerol cryo-protectant, reaching 30%. Diffraction data were collected at the European Synchrotron Radiation Facility (ESRF), using a helical collection strategy and maximum transmission of the ID29 beamline. Crystals are of space-group C2, diffracting at best to a maximum resolution of 4.0 Å. Data were integrated with STARANISO[27] to account for the anisotropy (Supplementary Table 1). The structure was solved with PHASER[28] using LACV-L$_{1-1750}$ (PDB code: 5AMQ)[13] as a model after removal of the endonuclease. There are two L protein complexes per asymmetric unit (Supplementary Fig. 1b). The initial map after molecular replacement revealed that the core of the L protein and bound

RNA were little changed but there was clear density for the endonuclease in a new position. In addition, there was extra density for the previously missing C-terminal domain. This density was improved by multi-crystal and non-crystallographic two-fold averaging using PHENIX[29]. Based on secondary structures predicted from an extensive multiple sequence alignment of *Peribunyaviridae* L proteins[30], it was possible to build an approximate model of much of the C-terminal domain (which is largely helical), except the CBD for which there is no density, including identification of the zinc-binding site co-ordinated by highly conserved cysteine and histidine residues. The two C-terminal domains from the two complexes in the asymmetric unit interact around a non-crystallographic two-fold axis in such a way that the extreme terminal helix of one packs against the C-terminal domain of the other, forming a domain swapped dimer. When the accurate structure of the C-terminal domain was obtained by cryo-EM, the X-ray model could be improved (Supplementary Fig. 1c, Supplementary Table 1). The two polymerases of the crystallographic asymmetric unit are very similar with an RMSD of 0.79 Å over 2009 Cα, although there is a slight difference in C-terminal domain orientation. They are also very similar to the cryo-EM structure at pre-initiation with RMSDs of, respectively, 3.022 and 2.951 Å over 2009 Cα. A significant difference concerns the C-terminal linker and last α-helix that form a domain swapped dimer interaction in the X-ray structure but folds back on the ZBD in the monomeric cryo-EM structure.

**Electron microscopy**. For cryo-EM experiments, LACV-L FL in complex with pre-annealed 3′ (1–16) and 5′ (9–16) vRNA at 0.2 mg/ml was mixed with 5′ (1–10) vRNA hook in a 1:2 molar ratio. UltraAuFoil grids 300 mesh, R 1.2/1.3 were negatively glow-discharged at 30 mA for 1 min. 3.5 μl of the sample was applied on the grids and excess solution was blotted away with a Vitrobot Mark IV (FEI) (blot time: 2 s, blot force: 1, 100% humidity, 20 °C), before plunge-freezing in liquid ethane. Grid screening and cryo-EM initial datasets were collected on a 200 kV Thermofischer Glacios microscope equipped with a Falcon II direct electron detector.

A high-quality cryo-EM grid pre-screened on a 200 kV Thermofischer Glacios microscope was used to collect data on a Thermofischer Titan Krios G3 operated at 300 kV equipped with a Gatan Bioquantum LS/967 energy filter (slit width of 20 eV) coupled to a Gatan K2 direct electron detector camera[31]. Automated data collection was performed with SerialEM using a beam-tilt data collection scheme[32], acquiring one image per hole from nine holes before moving the stage. Micrographs were recorded in super-resolution mode at a 165,000× magnification giving a pixel size of 0.4135 Å with defocus ranging from −0.8 to −3.5 μm. In total, 16,498 movies with 40 frames per movie were collected with a total exposure of 50 e−/Å[2] (Supplementary Table 2).

**Image processing**. Movie drift correction was performed in Motioncor2 using all frames, applying gain reference and cameras defect correction[33]. Images were binned twice, resulting in 0.826 Å/pixel size. Further initial image processing steps were performed in cryoSPARC v2.14.2[34]. CTF parameters were determined using "patch CTF estimation" on non-dose weighted micrographs. Realigned micrographs were then manually inspected using "Curate exposure" and low-quality micrographs were manually discarded for further image processing resulting in a curated 16,015 micrographs dataset (Supplementary Fig. 2c). LACV-L FL particles were then picked with "blob picker" using a circular particle diameter ranging from 90 to 150 Å, manually inspected and selected using "inspect particle picks", extracted from dose-weighted micrographs using a box size of 300 × 300 pixels[2], resulting in a 4,065,475 particles dataset. Successive 2D classifications were used to eliminate bad quality particles displaying poor structural features resulting in 2,279,573 particles suitable for further image processing (Supplementary Fig. 2b). Per particle CTF was calculated. The LACV-L FL initial 3D reconstruction was generated using "ab-initio reconstruction" in cryoSPARC using a small subset of particles. Subsequent steps were all performed in Relion 3.1[35,36]. The entire dataset was divided in four (~570,000 particles per subset) and subjected to 3D classification with coarse image-alignment sampling (7.5°) using a circular mask of 170 Å and 10 classes (labeled "1st 3D classification", Supplementary Fig. 2c). LACV-L FL classes displaying a stable C-terminal conformation were kept and merged for further classification resulting in 566,025 selected particles (dotted squared, green maps in Supplementary Fig. 2c). Selected particles were subjected to a 2nd run of 3D classification using finer angular sampling (3.7°) with a circular mask of 170 Å and was restricted to 10 classes (labeled "2nd 3D classification", Supplementary Fig. 2c). Particles from three 3D classes that display stable C-terminal regions without neighboring particles were selected to perform further high-resolution analysis, resulting in a 370,497 particle dataset. A previously obtained 3D class from the 2nd run of 3D classification was low-pass filtered to 15 Å, extended by 10 pixels with a soft edge of 5 pixels and used as a mask for a final 3rd run of 3D classification using finer angular sampling (1.8°) (labeled "3rd 3D classification", Supplementary Fig. 2c). Two classes (in dark blue, 59,152 particles) displayed a 10-base pair template–product RNA duplex in the LACV-L FL active site cavity and mimic an elongation state and one class (in cyan, 57,660 particles) displays a typical pre-initiation state with both 5′ 1–10 and 3′ 1–16/5′ 9–16 promoters bound. Both LACV-L FL pre-initiation and elongation-mimicking state subsets were submitted to 3D auto-refine using previous initial mask giving reconstructions at, respectively, 3.13 and 3.17 Å resolution using the FSC 0.143 cutoff criteria before post-processing. Masks to perform sharpening were generated using 3D refined

maps low-pass filtered at 10 Å, extended by 4 pixels with 8 pixels of soft-edge. Post-processing was done using an applied B-factor of −40 Å[2] and resulted in a map at 3.02 Å resolution for the elongation state and 3.06 Å resolution for the pre-initiation state using the FSC 0.143 cutoff criteria (Supplementary Figs. 2c, and 3a).

In order to deal with the high mobility and the small size of the CBD within LACV-L FL particles, the following advanced strategy was applied. 131,058 particles displaying a stable CBD (corresponding to its extreme position 1), originating from 4 out of the 10 classes from the third 3D classification (in orange in "third 3D classification-CBD view", Supplementary Fig. 2c) were submitted to 3D auto-refine in order to get the best global accuracy alignment. A mask excluding LACV-L$_{46-1751}$ residues, low-pass filtered to 10 Å, resampled and extended of 4 pixels with a soft edge of 6 pixels was then used for signal subtraction followed by particle re-centering on the mask center-of-mass. The resulting subtracted particles containing Endo$_{1-45}$-Mid-CBD densities were subjected to 3D masked auto-refine with local angular searches. The obtained map was sharpened with an applied B-factor of −90 Å[2] and resulted in a 3.54 Å resolution map according to the FSC 0.143 cutoff criteria.

A similar strategy was applied to deal with the ZBD flexibility. 287,363 particles displaying an ordered ZBD, originating from 7 out of the 10 classes from the third 3D classification (in red in "third 3D classification-ZBD view", Supplementary Fig. 2c) were submitted to 3D auto-refine in order to get the best global accuracy alignment. A mask excluding LACV-L$_{1-1751/1841-1983}$ residues, low-pass filtered to 10 Å, resampled and extended of 4 pixels with a soft edge of 6 pixels was used for signal subtraction followed by particle re-centering on the center-of-mass of this mask. Resulting subtracted particles containing mid-ZBD densities were classified without alignment in order to detect potential heterogeneity. The most stable subset containing 51,842 particles was subjected to 3D masked auto refine with local angular searches. The resulting map was post processed with an applied B-factor of −83 Å[2], which resulted in a 3.49 Å resolution map according to the FSC 0.143 cutoff criteria.

For each final map, local resolution variations were estimated in Relion 3.1 (Supplementary Fig. 3). The 3D variability analysis was performed in cryoSPARC filtering resolution to 4 Å and using three modes.

**Model building in the cryo-EM maps**. All the cryo-EM maps, namely pre-initiation map, elongation-mimicking map, CBD-mid domain map, and ZBD-mid domain map were superimposed using Chimera[37] previous to model building. The partial model determined in the 4.0 Å X-ray structure was used as a starting point to manually build into the cryo-EM maps using COOT[38]. The map chosen for model building was the one corresponding to the best resolution in the region built. The pre-initiation and elongation-mimicking maps were used to build the LACV-L core, the endonuclease domain, the mid domain and the β-hairpin strut of the ZBD. The CBD-mid domain map was used to build the CBD. The ZBD-mid domain map was used to build the ZBD devoid of its β-hairpin strut. The sequence of the RNA duplex visible in the active site cavity at elongation state was deduced based on purines and pyrimidines, clearly visible in the density. As the 3′ and 5′ vRNA incubated respectively contain 16 and 10 nucleotides, 6 nucleotides of the vRNA should be present in the product exit tunnel and none in the template exit tunnel. Blurred density is visualized in the product exit tunnel due to the large flexibility of the nucleotides. Blurred RNA density is also visible in the template exit tunnel suggesting that some particles have encapsidated the RNA in different positions than the one shown in Figs. 1c and 5d. After initial manual building in COOT, the models were iteratively improved by Phenix-real space refinement[39] and manual building in COOT. Validation was performed using the Phenix validation tool, and model resolution was estimated at the 0.5 Fourier shell correlation (FSC) cutoff. Figures were generated using ChimeraX[40].

**Multiple alignment**. Six sequences of major Peribunyaviruses (L_BUNYW for Bunyamwera virus, L_SBVBG for Schmallenberg virus, L_MCAV for Macaua virus, L_WBV for Wolkberg virus, L-Oya for Oya virus) were aligned using MUSCLE[41] and presented with ESPript[42].

**Reporting summary**. Further information on research design is available in the Nature Research Reporting Summary linked to this article.

## Data availability

Coordinates and structure factor have been deposited in the Protein Data Bank and the Electron Microscopy Data Bank with the accession codes: LACV-L pre-initiation complex (X-ray crystallography) PDB 6Z6B, LACV-L pre-initiation complex PDB 6Z6G EMDB EMD-11093, LACV-L elongation complex PDB 6Z8K EMDB EMD-11118, LACV-L CBD and mid domain map EMDB EMD-11095, LACV-L ZBD and mid domain map EMDB EMD-11107. Other data are available from the corresponding authors upon reasonable request.

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

## Acknowledgements

We thank Karine Huard, Angélique Fraudeau, and Alice Aubert for technical advices on expression and purification; Ambroise Desfosses, Leandro Estrozi, and Alister Burt for discussion on image processing; Aymeric Peuch for help with the usage of the EM computing cluster; Dominique Housset for general discussion on the project and Irina Gutsche for support. This work used the platforms of the Grenoble Instruct-ERIC center (ISBG; UMS 3518 CNRS-CEA-UGA-EMBL) within the Grenoble Partnership for Structural Biology (PSB), supported by FRISBI (ANR-10-INBS-05-02), and GRAL, financed within the University Grenoble Alpes graduate school (Ecoles Universitaires de Recherche) CBH-EUR-GS (ANR-17-EURE-0003). The electron microscope facility is supported by the Auvergne-Rhône-Alpes Region, the Fondation Recherche Médicale (FRM), the fonds FEDER, and the GIS-Infrastructures en Biologie Santé et Agronomie (IBISA). We acknowledge the European Synchrotron Radiation Facility (ESRF) for provision of beam time on CM01. We thank all platform staff that enabled us to perform these analyses. This work was supported by the IDEX IRS G7H-IRS17H50 and the ANR-19-CE11-0024-02 to H.M.

## Author contributions

B.A., P.G., and J.R. expressed and purified LACV-L FL. P.G. crystallized LACV-L FL. S.C. P.G. and J.R. solved the X-ray structure. S.C. built an initial model based on X-ray crystallography data. B.A. prepared cryo-EM grids. B.A. and H.M. collected cryo-EM data on a Thermofischer Glacios EM thanks to advices and training from G.S. who set up and maintains the IBS EM platform. G.E. set up SerialEM data collection scheme and collected the high-resolution cryo-EM dataset on the Thermofischer Krios EM at ESRF (CM01). B.A. performed advanced cryo-EM image processing and 3D reconstructions. H.M., B.A., and S.C. built the models based on the cryo-EM maps and performed structural analysis. H.M. and G.S. co-supervise B.A.; J.R. and S.C. co-supervised P.G. The project was conceived by H.M. and S.C. This project used funding obtained by H.M., G. S., and S.C. The manuscript was written by H.M. and B.A. with input from all authors.

## Competing interests

The authors declare no competing interests.
