## [Peer Review File · Nature Communications]

Reviewer #1 (Remarks to the Author):

Arragain et al have used X-ray crystallography and cryo-EM to obtain a high-resolution structure of the La Crosse virus L protein. This approach enabled them to build a full-length structure of the polymerase core and the flexible cap binding and endonuclease domains. In addition, the authors were able to obtain a structure of the L protein bound to RNA in an elongation complex. The work also advances our understanding of this enzyme complex by showing new orientations of the flexible domains, suggesting that these move around during cap snatching. The identification of the flexible zinc binding domain is also very intriguing. Overall the work is of high quality, the interpretation of the structures very sensible and I believe that this work significantly advances our understanding of RNA virus RNA polymerases and, in particular, our understanding of bunyavirus RNA synthesis. I highly recommend this manuscript's publication and just have a couple of minor comments that the authors could discuss.

1. The authors speculate that the zinc binding domain is a host factor binding platform and make a comparison with the 627-domain of the influenza virus RNA polymerase, which is involved in host adaptation and viral replication. The NCBI lists at least 84 La Crosse virus L protein sequences. Did the authors check if mutations are present in the zinc binding domain that could reveal more about its function?

2. The authors say that the active site is slightly more open than normal (line 210). What post-incorporation complex are they comparing the LACV polymerase too? I presume the influenza A virus one, but it would be good to have this specified and perhaps even quantified since this structure is such an interesting intermediate between incorporation and translocation. In addition, do the authors assume that some of the energy that will power translocation is stored in this open conformation?

Reviewer #2 (Remarks to the Author):

Arragain et al. reports the X-ray crystallographic cryo-EM structures of the polymerase of La Crosse Virus (LACV). Only a single existing structure of a Bunyavirales polymerase exists and is missing crucial information for the C-terminal portion of the enzyme which contains the cap-binding domain (CBD) and a zinc-binding domain (ZBD). Bunyavirales members utilize a cap-snatching mechanism to grab a cellular mRNA and utilize an N-terminal endonuclease domain to cleave the mRNA, leaving a capped template that serves as a primer for transcription and replication. The length of the truncated RNA differs based on virus species and it's unknown whether L interacts with cellular factors like RNA polymerase II to grab the mRNA as has been shown in influenza. Here the authors present both the x-ray crystallographic and cryo-EM structures of full length LACV-L protein in both an apo state and bound to a duplexed RNA template that mimics the template and product strand. Overall this is an important structure that reveals the full structure of an L protein from this order of viruses. However, the manuscript could use some editing in order to polish the text to make it easier for the reader to understand.

1) on line 51 it should read RNA polymerase II instead of just polymerase II.

2) Interestingly the conformation of the endonuclease domain differs from the previously published structure of a C-terminally truncated (1-1750) LACV-L construct by ~180 degrees. This was somewhat hard to visualize in Supplemental Figure 4 and the authors might consider adding labels to helices to better visualize this difference.

3) Why this difference in position of the endonuclease is not addressed by the authors. Might it be due to the presence of the full length construct forming additional interactions between endonuclease and the CBD?

4) In general I found many figures to be very cluttered and difficult to focus on. The previously missing C-terminal portion of L projects away from the body of the polymerase with the CBD

interacting with the endonuclease domain. The CBD is flexible as shown in Figure 3C but the authors might consider using different colors so as not to be confusing with the N to C terminal coloring in Figure 3D & E.

5) This flexibility is compared to existing structures of the influenza virus polymerase but the range of motion identified here (~35 degrees) doesn't encompass the states seen in influenza that correlate with bringing the cellular RNA first to the endonuclease for cleavage and then to the template entry site.

6) The reference on line 158 to Supplemental Figure 10 needs to be corrected.

7) The authors claim that alpha helix 91 folds back into the hydrophobic pocket of the ZBD. It suggests a stable interaction although the electron density for that helix is largely missing at a standard threshold, indicating it is highly flexible and not very stable.

8) Surprisingly the authors were able to identify a subset of particles with duplexed RNA in the active site cavity which the authors suggest it represents an elongation state. It is consistent with the path of the RNA from the 5' RNA promoter bound in the 5' stem-loop pocket. The authors were able to identify differences in the position of the priming loop and a portion of the mid-domain that it interacts with in the elongation state compared to the pre-initiation state. On lines 248-251 a reference is made to residue Y1696 forcing separation of the two strands but I don't see any labeling of that residue in the referenced figure, 4b. Might the authors have meant 4c?

Reviewer #3 (Remarks to the Author):

La Crosse virus (LACV) is a significant pathogen of Bunyavirales; however, no effective treatment and vaccine are available. The viral RNA-dependent RNA polymerase of LACV is responsible for both replication and transcription of the viral genome. Arragain et al. described both the crystal and cryo-EM structures of full-length LACV polymerase in two different functionally states, pre-initiation and elongation. In particular, compared to the previously reported structure of LACV-L(1-1750) that lacks C-terminal domains (CTD), Arragain et al. reported the structure of the full-length of LACV-L, including the model of the C-terminal domains and revealed the changes between two captured functional states. Arragain et al. also demonstrated that the priming loop, lid domain, and C-terminal region are involved during RNA synthesis.

Overall, it is a nice paper with robust experimental data. Arragain et. al. used a combination of x-ray crystallography and cryo-EM to determine the structures of the full-length LACV polymerase with two functional states. The Arragain et al. revealed the difference between the pre-initiation and elongation states, highlighted the structural features of the cap-binding domain (CBD, residues 1842-1977) and zinc-binding domain (ZBD, residues 2026-2263) within the CTD.

However, there are some issues that need to be addressed:

1. Due to a large amount of information, in the data statistics table, the author should list the components for each structure, including the sequence of 3' template, 3' 1-16, 5' hook, ENDO CORE, and the range of protein residues.

2. Page 5 "adjusted version of the protocol described previously". Please briefly highlight and describe the difference compared to the protocol by Gerlach et al. (2013).

3. It is interesting to see not only the pre-initiation state of complex but also an elongation state with a bound product-template duplex. The 5' promoter 1-10 and 3' promoter 7-16 is not a perfect match (with one UG mismatch), but very close. The captured state is without the incoming NTP and longer template at the 5' end. So it is curious to see what are the distances between the active site residues and the U16? Is this close enough to catalyze the reaction? Also, since it is unexpected, could this be an inhibition state rather than an elongation state?

4. Please make the residue numbering consistent throughout the text. For example, on pages 5

and 6, LACV-L 1-1750 vs and the polymerase core (residues 186-1751). It is not apparent that the C-terminal region (residues 1752-2263) includes the mid-domain (residues 1752-1841 and 1978-2025), C-terminal domain (CBD, residues 1842-1977), and zinc-binding domain (ZBD, residues 2026-2263), and the β -hairpin strut (residues 2084-2102) is part of ZBD. Please rearrange the text and make it easy to appreciate the relationships among domains.

5. Page 7: Fig.2d and Fig.2e are missing? Should be Fig.3d and Fig.3e? Not clear about the extreme position 1 and 2. The color is confusing. The color of the ribbons in Fig. 3a is also gold and orange. Do the gold and red in Fig.3c and Fig.3d represent the entire CBD or part of the CBD? What regions are they comparing to? Please show the comparison with one region (such as endonuclease domain) is fixed.

6. "Its two sub-domains of equivalent size surround a metal ion that is coordinated by four residues highly conserved amongst peribunyaviruses: C2064, H2169, D2178 and H2182, suggesting that it is a zinc ion. The overall topology of the ZBD has not been previously observed according to a DALI search." It is interesting as a ZBD; it doesn't have a structural homology. Does any other related polymerase have a similar ZBD? Or is this unique in LACV.

7. Figure 1. Too many colors in Figure 1 make it hard to recognize the details of the structures. Also, please do not switch the color for the same component. For example, the template 5'vRNA, and the product 3'vRNA should keep with consistent color within the same figure. The name of the sequences in the figure should be labeled clearly and consistent with the main text, including 3' vRNA, 5' vRNA, template, and product. The figure legends should contain more details to illustrate the figures clearly, such as the color and residue numbers in Figure 1a.

8. The sequence alignments should include the protein ID (such as the UniProt #) in Figure 3b and Supplementary Figure 10.

9. Supplementary figure 9: the comparison between LACV and influenza polymerase priming loop at elongation. It seems that the priming loop in LACV is not as flexible as influenza polymerase. Because the priming loop is supposed to be critical in the initiation stage, but not in the elongation stage, does this mean the priming loop in LACV is in a state between initiation and elongation?

10. There are two polymerases in the asymmetric unit of the crystal structure. Are they identical? Please also compare them with the cryo-EM structure at the pre-initiation state.

11. Page 16. "Before freezing and storing at -80°C, LACV-L bound to 3' (1-16) and 5' (9-16) vRNA were mixed with 5' (1-10) vRNA hook in a 1:2 molar ratio." Why not incubate them at the same time? Also, why choose 1:2 ratio? In addition, "The 5' (1-10) vRNA end was later soaked into crystals." Using the same ratio?

We thank you for your positive appreciation of and constructive comments on our manuscript which focuses on the structure of the full-length La Crosse virus polymerase. Please find below our point-by-point responses.

Reviewer #1 (Remarks to the Author):

1. *The authors speculate that the zinc binding domain is a host factor binding platform and make a comparison with the 627-domain of the influenza virus RNA polymerase, which is involved in host adaptation and viral replication. The NCBI lists at least 84 La Crosse virus L protein sequences. Did the authors check if mutations are present in the zinc binding domain that could reveal more about its function?*

The structure of LACV zinc-binding domain adopts a fold not previously observed and its function is currently unknown. In order to possibly identify residues that systematically differ from one strain to another which might reflect host specificity, we performed an alignment of all sequences from La Crosse virus polymerases present in the NCBI (36 sequences in the non-redundant protein database). The only residues that differ from one strain to another correspond to conservative mutations of surface residues. We carefully checked for host specificity and/or geographic specificity of the mutations but this analysis did not reveal any specific pattern. This shows that, contrary to influenza PB2 that displays a glutamate or a lysine at position 627 which reflects host specificity, it is not possible to identify a non-conservative mutation reflecting specific host or geographic adaptation in LACV-L ZBD. In the text, we therefore now mention this (lines 307-309):

“It should however be noticed that, contrary to influenza 627-domain, it is not possible to identify a non-conservative mutation of a residue in LACV-L ZBD that would differentiate one LACV strain to another and might reflect host specificity.”

We also did the same analysis with the 84 sequences of L proteins from the *Peribunyaviridae* family present in RefSeq (out of which 71 correspond to full length protein sequences). These sequences correspond to different viruses and display significant divergence that prevents to identify any residue that would reflect insect host specificity. It however identifies that the four amino acids directly involved in zinc binding are conserved in all *Peribunyaviridae* polymerase sequences, clearly showing that ion binding is a conserved function. This is now indicated lines 180-181:

“These four residues occur unchanged in all the 84 *Peribunyaviridae* sequences deposited in the NCBI, indicating that ion binding is a conserved feature in this viral family.”

2. *The authors say that the active site is slightly more open than normal (line 210). What post-incorporation complex are they comparing the LACV polymerase too? I presume the influenza A virus one, but it would be good to have this specified and perhaps even quantified since this*

structure is such an interesting intermediate between incorporation and translocation. In addition, do the authors assume that some of the energy that will power translocation is stored in this open conformation?

We agree with Reviewer 1 that the observed state appears to mimic an intermediate between incorporation and translocation, and now discuss this more explicitly in the text. The influenza polymerase structures that we use for comparison are now mentioned. We have added a Supplementary Figure 6 that compares the LACV elongation-like structure with:

- influenza polymerase in the post-incorporation pre-translocation state (PDB 6SZU) that displays a pyrophosphate and two magnesium ions in catalytic positions.

- influenza polymerase in the post-incorporation post-translocation state (PDB 6QCT) that contains a magnesium in an inactive open state.

As requested by Reviewer 3, interaction distances between RNA, active site residues and magnesium ions have been added.

To respond to the second question of Reviewer 1, we think it difficult to speculate from a structure about energy storage to power. Indeed, the structure observed here only mimics an elongation post-incorporation state. It does not result from polymerase catalytic activity but by fortuitous internalization of an RNA-duplex, as clearly mentioned in the text. As a result, it may not represent a true intermediate, even if some active site elements appear to adopt a post-incorporation conformation due to the presence of the 10-bp double stranded RNA (the motif B for example). The particular configuration we see may reflect the absence of the pyrophosphate normally present after nucleotide incorporation. This may reflect a true intermediate where the pyrophosphate has fully dissociated, but we thus prefer to be cautious and avoid over-interpretation of the data. We thus restrict ourselves to the description of the structure explaining how it mimics a post-incorporation pre-translocation stage in the result. We compare with influenza L post-incorporation/pre-translocation and post-incorporation/post-translocation in the discussion.

The modified text in the result section (lines 233-246) is the following:

“The polymerase conformation mimics a post-incorporation, pre-translocation elongation step in which an incoming nucleotide would just have been incorporated into the product. During the nucleotide addition cycle, viral polymerase active sites undergo small structural changes that enable NTP-binding, NTP-incorporation and subsequent RNA translocation²⁰. Whereas a particular configuration of (i) the nucleotide to be incorporated, (ii) the product RNA to be elongated, (iii) two magnesium ions and (iv) the three aspartic acids of motifs A and C is necessary for catalysis of the phosphoryl transfer reaction, the organisation of the active site changes subsequently²⁰. Such post-incorporation structural changes are visible in LACV elongation mimicking stage. Perhaps related to the fact that there is no pyrophosphate (since there was no reaction), the two magnesium ions that control nucleotide addition are not present in the catalytic configuration. Instead, a presumed Mg²⁺ ion, coordinated by residues D1188 (motif C), E1237 (motif E) and the carbonyl oxygen of A1059 (motif A) is present, typical of the inactive open state of the polymerase active site (**Fig. 5b, Supplementary Fig. 6a**).”

The modified discussion is (lines 349-362):

The active sites of LACV and influenza polymerases in presence of RNA are remarkably similar (**Supplementary Fig. 6**). Previous studies of influenza polymerase have depicted the post-incorporation pre-translocation (**Supplementary Fig. 6b**)²³ and post-incorporation post-translocation (**Supplementary Fig. 6c**)¹² states. LACV-L FL bound to duplex

RNA appears to mimic an intermediate between these two states as it displays a product nucleotide at the +1 position (as if it had been newly incorporated) with a magnesium ion and the catalytic aspartic acids in an inactive open state conformation (**Supplementary Fig. 6a**). Such an intermediate resembles the equivalent state in poliovirus polymerase (PDB: 3OL9)²⁶ although the observed movements of the active site between pre/post translocations are restricted to side chains movements in LACV/influenza structures, whereas they involve larger movements of motifs in poliovirus polymerase. It should however be noticed that the observed state in LACV-L was obtained by internalization of a double-stranded RNA and might not exactly reflect a true elongation intermediate notably because of the absence of pyrophosphate.

Reviewer #2 (Remarks to the Author):

1) on line 51 it should read RNA polymerase II instead of just polymerase II.

This has been corrected (now corresponds to line 53-54).

2) Interestingly the conformation of the endonuclease domain differs from the previously published structure of a C-terminally truncated (1-1750) LACV-L construct by ~180 degrees. This was somewhat hard to visualize in Supplemental Figure 4 and the authors might consider adding labels to helices to better visualize this difference.

Labels have been added on helices of Supplementary figure 4b in order to facilitate the visualization of the endonuclease rotation.

3) Why this difference in position of the endonuclease is not addressed by the authors. Might it be due to the presence of the full-length construct forming additional interactions between endonuclease and the CBD?

We have added a paragraph in the main results and the main Figure 2 describing the interaction between the endonuclease and the rest of the polymerase. In addition, we have re-written a paragraph of the discussion, in which we compare the endonuclease position in LACV-L FL and LACV-L₁₋₁₇₅₀. We also suggest potential necessary conformational changes of the endonuclease in order to enable transcription initiation to occur and in order to avoid degradation of the product at late elongation stage.

The text added in the result is the following (lines 137-149):

“The endonuclease is held in place by hydrophobic interactions with a large number of residues from different domains (**Fig. 2a**). The N-terminus of the endonuclease and its un-cleaved TEV cleavage site are buried between the thumb ring (residues 1716, 1717, 1720) and the residue 714 of the core lobe β -hairpin (**Fig. 2b and a**). The α -helices 6 and 7 of the endonuclease face the palm domain (α -helix 38) and the finger domain (α -helix 30) (**Fig. 2a and c**). Hydrophobic interactions are depicted between residues 46-53 (corresponding to the

linker between α -helices 2 and 3) and the mid domain (residue 1813 and residues of the α -helix 76). Residues 137-159 of the endonuclease (corresponding to the β -strand 4, the α -helix 5 and the linker between the α -helices 5 and 6) interact with the residues 1610-1614 of the thumb ring domain, the α -helix 68 of the lid domain, the α -helix 71 of the thumb-ring domain, the mid-thumb-ring linker (residues 1741-1751) and the α -helix 72 of the mid domain (**Fig. 2d**). Finally, the endonuclease interacts with the CBD (**Fig. 2a**), through interactions that change depending on the CBD position (described in the paragraph below)."

Description of the CBD-endonuclease interaction is described here (lines 168-174):

In the extreme position 1, residues 12-28 and 175-178 of the endonuclease domain interact with residues E1894, R1901, R1930, 1950-1960 of the CBD mainly through electrostatic interactions (**Fig.3d**). In the extreme position 2, the contacts between the CBD and the rest of the polymerase are rather sparse, explaining its instability (**Fig.3e**). The only interactions are mediated by the loop 1932-1936 of the CBD that is proximal to C20 of the endonuclease domain, and the residue E1894 of the CBD that is close to the H75 of the endonuclease domain (**Fig.3e**).

The text added in the discussion is the following (lines 316-321 and 329-335):

"In both conformations, the endonuclease is stabilised in position by making hydrophobic interactions with the thumb ring, core lobe, mid, palm, finger and cap-binding domains (**Fig.2**). Its orientation however differs by around 180 degrees compared to LACV-L₁₋₁₇₅₀ (**Supplementary Fig. 4**) where the endonuclease protrudes away from the polymerase core, making only few interactions with it."

"However, the observed location of the endonuclease prevents rotation of the CBD into a position where it can direct the capped primer into the active site for transcription initiation (**Supplementary Fig. 8a**), so a relative re-positioning of the two domains is expected. In addition, the endonuclease active site is located close to the product exit (**Supplementary Fig. 8a**). In order to prevent degradation of the product when it exits the polymerase, one can speculate that a large movement of the endonuclease may be necessary at a late elongation stage."

4) *In general I found many figures to be very cluttered and difficult to focus on. The previously missing C-terminal portion of L projects away from the body of the polymerase with the CBD interacting with the endonuclease domain. The CBD is flexible as shown in Figure 3C but the authors might consider using different colors so as not to be confusing with the N to C terminal coloring in Figure 3D & E.*

In order to help the reader focusing on the figures and in order to answer Reviewer 2 point 4 and Reviewer 3 point 7, we have reduced the coloring to the minimum, keeping only colors when the domains are mentioned in the text and are therefore necessary for text comprehension. As a result, several features like the clamp, the arch, the vRBL, the α -ribbon, the California insertion and the finger nodes are not colored anymore but reattached to the domain to which they belong.

Coloring of Figure 3c have been modified in order to avoid confusion with coloring of figure 3a, b, d and e.

5) This flexibility is compared to existing structures of the influenza virus polymerase but the range of motion identified here (~35 degrees) doesn't encompass the states seen in influenza that correlate with bringing the cellular RNA first to the endonuclease for cleavage and then to the template entry site.

We agree with Reviewer 2 that the range of motion observed does not correspond to the movement observed in influenza that correlate with bringing the cellular RNA first to the endonuclease for cleavage and then to the template entry site. We can only hypothesize about the function of the observed conformations called "LACV-L conformation 1" and "LACV-L conformation 2". If LACV-L conformation 2 corresponds to a state in which capped RNA is cleaved, it is clear that we are missing the state in which capped RNA is brought to the template entry site. Other structures in complex with capped RNA will be necessary to visualize more states and fully understand the mechanisms underlying transcription initiation. We have amended the discussion to clarify this. We have removed the previous Supplementary Figure 7 and introduced a new Supplementary Figure 8 that describes this. We have also modified the text as follow (lines 324-346):

"These conformations are likely to correspond to functional states. LACV-L conformation 1 may correspond to a pre-initiation transcription state compatible with the binding of cellular capped RNA, as its CBD is exposed towards the exterior (**Supplementary Fig. 8a**). In LACV-L conformation 2, the cap-binding site is closer to the endonuclease and we speculate that in this conformation the RNA can be cleaved by the endonuclease (**Supplementary Fig. 8a**). However, the observed location of the endonuclease prevents rotation of the CBD into a position where it can direct the capped primer into the active site for transcription initiation (**Supplementary Fig. 8a**), so a relative re-positioning of the two domains is expected. In addition, the endonuclease active site is located close to the product exit (**Supplementary Fig. 8a**). In order to prevent degradation of the product when it exits the polymerase, one can speculate that a large movement of the endonuclease may be necessary at a late elongation stage.

The position of both the endonuclease and the CBD in LACV-L conformation 1 and 2 significantly differ from the observed disposition of these domains in all the functional states described for influenza polymerase (**Supplementary Fig. 8b**). They also diverge from the recently described structures of full-length L protein from Machupo virus (*Arenaviridae* family)¹⁷ and Severe Fever with Thrombocytopenia Syndrome virus L (*Phenuiviridae* family)^{24,25} published while this manuscript was under review (**Supplementary Fig. 8c,d**). However, without being sure that these latter structures represent functionally active states, it is too early to draw any conclusions on differences in mechanism between these various viral polymerases. The insight provided here on LACV-L constitutes a basis to address in more detail the exact mechanisms underlying *Peribunyaviridae* transcription initiation in the future, notably by determining structures in complex with capped RNA."

6) The reference on line 158 to Supplemental Figure 10 needs to be corrected.

This has been done.

7) *The authors claim that alpha helix 91 folds back into the hydrophobic pocket of the ZBD. It suggests a stable interaction although the electron density for that helix is largely missing at a standard threshold, indicating it is highly flexible and not very stable.*

We managed to obtain a better map of the ZBD in which the α -helix 91 is better defined and we deposited this improved map in EMDDB under the accession number EMD-11107. We however acknowledge that the α -helix 91 remains less defined than some other parts of the ZBD. We have thus modified the text to clearly mention that the α -helix 91 is more flexible than other parts (lines 197-200):

“In the cryo-EM map, the polymerase is monomeric and some density present at low threshold suggests that the α -helix 91 may fold back into the same hydrophobic pocket of the ZBD, although the binding might be rather labile (Fig.4a).”

8) *Surprisingly the authors were able to identify a subset of particles with duplexed RNA in the active site cavity which the authors suggest it represents an elongation state. It is consistent with the path of the RNA from the 5' RNA promoter bound in the 5' stem-loop pocket. The authors were able to identify differences in the position of the priming loop and a portion of the mid-domain that it interacts with in the elongation state compared to the pre-initiation state. On lines 248-251 a reference is made to residue Y1696 forcing separation of the two strands but I don't see any labeling of that residue in the referenced figure, 4b. Might the authors have meant 4c?*

Yes, Reviewer 2 is right and this was corrected. (Due to the introduction of a new Figure 2, it has been corrected to Fig. 5c, line 277).

Reviewer #3 (Remarks to the Author):

1. *Due to a large amount of information, in the data statistics table, the author should list the components for each structure, including the sequence of 3' template, 3' 1-16, 5' hook, ENDO CORE, and the range of protein residues.*

The list of the domains built into each map is specified and the residues range indicated as requested. The sequence of the RNA present has been added (residues present but not built due to their too high flexibility are shown in italics, this is mentioned in the statistics table).

2. *Page 5 “adjusted version of the protocol described previously”. Please briefly highlight and describe the difference compared to the protocol by Gerlach et al. (2015).*

The protocol is fully described in the material and methods. In addition, we now highlight the differences compared to the protocol by Gerlach et al, 2015 in the main text (lines 99-106):

“LACV-L FL was expressed in insect cells and purified to homogeneity based on the protocol described in Gerlach *et al.*¹³ (**Supplementary Fig. 1a**). Slight modifications were however necessary in order to stabilize LACV-L FL, in particular the addition of nucleotides 1 to 16 of the 3' vRNA (3'OH-UCAUCACAUGAUGGUU) and complementary 8-mer corresponding to the nucleotides 9 to 16 of the 5'vRNA (5'OH-GCUACCAA) prior to the decrease to 150 mM of NaCl concentration in the buffer. LACV-L was then further stabilized by the addition of the first 10 nucleotides of the 5' vRNA (5'pAGUAGUGUGC), that were added by crystal soaking or just before cryo-EM grid freezing.”

3. It is interesting to see not only the pre-initiation state of complex but also an elongation state with a bound product-template duplex. The 5' promoter 1-10 and 3' promoter 7-16 is not a perfect match (with one UG mismatch), but very close. The captured state is without the incoming NTP and longer template at the 5' end. So it is curious to see what are the distances between the active site residues and the U16? Is this close enough to catalyze the reaction? Also, since it is unexpected, could this be an inhibition state rather than an elongation state?

As suggested, we now display distances between U16 and the active site in the new Supplementary Figure 6. As mentioned in the initial version of the manuscript and in the new version proposed here, the position of the catalytic residues and the magnesium ion do not exactly correspond to a catalytic state. They rather mimic another relevant polymerase state: the post-incorporation state (as suggested in Reviewer 1 point 2). We have now made clear that the polymerase state mimics an intermediate position between incorporation and translocation. This is exemplified by the comparison with Influenza polymerase in post-incorporation pre-translocation state (Supplementary Fig. 6b) and in post-incorporation post-translocation state (Supplementary. Fig 6c).

4. Please make the residue numbering consistent throughout the text. For example, on pages 5 and 6, LACV-L 1-1750 vs and the polymerase core (residues 186-1751). It is not apparent that the C-terminal region (residues 1752-2263) includes the mid-domain (residues 1752-1841 and 1978-2025), C-terminal domain (CBD, residues 1842-1977), and zinc-binding domain (ZBD, residues 2026-2263), and the β -hairpin strut (residues 2084-2102) is part of ZBD. Please rearrange the text and make it easy to appreciate the relationships among domains.

LACV-L 1-1750 is the construct previously crystallized. It does not correspond to the core that is the stable part of the full-length polymerase (residues 186-1751). As requested by Reviewer 3, we have arranged the text to make clear that the C-terminal region contains the mid domain, the cap-binding domain and the ZBD. We also clearly state that the β -hairpin strut (residues 2084-2102) is part of ZBD. The modified text is the following (lines 130-135):

“The previously unobserved C-terminal region (1752-2263) protrudes away from the core and forms an elongated arc-shaped structure that includes the mid-domain (residues 1752-1841 and 1978-2025), the CBD (residues 1842-1977), and the zinc-binding domain (ZBD, residues 2026-2263) (**Fig.1a and b**). The C-terminal region is supported and stabilized by a β -hairpin strut (residues 2084-2102) that emerges from the ZBD domain and bridges the entire C-terminal region to the core (**Fig.1b**).”

This is also shown with a schematic on Figure 1a. For clarity, we have added the label “C-terminal region” 1752-2263 on Fig. 1a.

5. *Page 7: Fig.2d and Fig.2e are missing? Should be Fig.3d and Fig.3e? Not clear about the extreme position 1 and 2. The color is confusing. The color of the ribbons in Fig. 3a is also gold and orange. Do the gold and red in Fig.3c and Fig.3d represent the entire CBD or part of the CBD? What regions are they comparing to? Please show the comparison with one region (such as endonuclease domain) is fixed.*

Fig.2d, Fig.2e indicated in the text were wrongly labeled. The correction has been made and we now refer to Fig.3d, Fig.3e.

We now choose different colors to describe the two conformations of the entire CBD (Fig. 3c) in order to avoid confusion. For Figure 3c, we compare the CBD position to the core and the endonuclease that have stable fixed position. This is now stated in the legend of Fig.3 lines 718-719:

“CBD movement is compared to LACV-L core and endonuclease that adopt stable positions.”

Colors of Fig 3. a,b,d and e are consistent. As written on the figure and in the legend, the secondary structures conserved with other sNSV CBD are shown in red and yellow. LACV CBD insertion are shown in orange.

6. *“Its two sub-domains of equivalent size surround a metal ion that is coordinated by four residues highly conserved amongst peribunyaviruses: C2064, H2169, D2178 and H2182, suggesting that it is a zinc ion. The overall topology of the ZBD has not been previously observed according to a DALI search.” It is interesting as a ZBD; it doesn’t have a structural homology. Does any other related polymerase have a similar ZBD? Or is this unique in LACV.*

As mentioned by Reviewer 3 and indicated in the text, the overall fold has not been previously observed in general in the PDB and therefore not in other related polymerases up to now. Binding of zinc ions by polymerases have however been reported in several other polymerases. This is now indicated lines 181-182:

“Whereas zinc ion binding by viral polymerases is rather common¹⁵⁻¹⁷, the overall topology of the LACV ZBD has not been previously observed according to a DALI search¹⁸.”

7. *Figure 1. Too many colors in Figure 1 make it hard to recognize the details of the structures. Also, please do not switch the color for the same component. For example, the template 5’vRNA, and the product 3’vRNA should keep with consistent color within the same figure. The name of the sequences in the figure should be labeled clearly and consistent with the main text, including 3’ vRNA, 5’ vRNA, template, and product. The figure legends should contain more details to illustrate the figures clearly, such as the color and residue numbers in Figure 1a.*

In order to help the reader focusing on the figures and in order to answer Reviewer 2 point 4 and Reviewer 3 point 7, we have reduced the coloring to the minimum, keeping only colors when the domain is mentioned in the text and is therefore necessary for text comprehension. As a result, several features like the clamp, the arch, the vRBL, the alpha-ribbon, the California insertion and the finger nodes are not colored anymore but reattached to the domain to which they belong.

We now keep consistent coloring for the same component, in particular the 5' and 3' RNA as requested by Reviewer 3. More generally we aim at being consistent in coloring as much as possible in all the figures. We are now clearly labeling 3'RNA, 5'RNA, template and product in Figure 1c where we give the RNA sequence.

8. The sequence alignments should include the protein ID (such as the UniProt #) in Figure 3b and Supplementary Figure 10.

This has been added. (Supplementary Figure 10 is now Supplementary Data 1).

9. Supplementary figure 9: the comparison between LACV and influenza polymerase priming loop at elongation. It seems that the priming loop in LACV is not as flexible as influenza polymerase. Because the priming loop is supposed to be critical in the initiation stage, but not in the elongation stage, does this mean the priming loop in LACV is in a state between initiation and elongation?

At the initiation stage the priming loop protrudes towards the active site in order to stabilize the replication initiation state. In the pre-initiation state that we observe, the primer loop protrudes towards the active site but is disordered, probably due to the absence of RNA and nucleotides. This is classic in pre-initiation state of many viral polymerase structures.

At elongation, the priming loop extrudes from the active site to enable binding of the nascent double-stranded RNA. In LACV elongation-mimicking state the priming loop is fully extruded, in a position compatible with the binding of a 10-base pair double stranded RNA. It thus corresponds to an elongation position. The priming loop of LACV may be stabilized while influenza polymerase priming loop is not for two reasons:

-LACV-L priming loop contains 20-amino acids whereas Influenza polymerase priming loop contains 35 amino acids. It is therefore logical that influenza polymerase priming loop protrudes more towards the exterior and is more flexible.

-LACV-L priming loop is stabilized in the elongation-mimicking stage by the helix-72 of the mid/thumb ring linker, a feature absent in influenza polymerase.

We therefore think that the priming loop is at elongation and that influenza and LACV polymerase priming loops extrude in different ways, highlighting a divergence between these two viral families. This is discussed lines 259-271 and 368-377:

“In the pre-initiation structure, this loop protrudes towards the active site but is disordered probably due to the absence of RNA and nucleotides (**Fig. 6a**). As part of the initiation to elongation transition, it extrudes from the active site via the template exit tunnel, thereby freeing space for the ten-base-pair RNA to fit in the active site chamber (**Fig. 6b**). The fully ordered and extruded priming loop is located on the surface of the thumb ring and lid domains (**Fig. 6b**) with which it interacts mainly through hydrophobic contacts involving

residues V1572, Y1576, A1751, M1753, N1658, L1661 (**Fig. 6d**). Interestingly, the priming loop movement is coupled with the reorganisation of mid-domain residues 1752 to 1761 from an α -helix to an extended loop. This results in an extension of the mid-thumb-ring linker from residues 1741-1751 at pre-initiation to 1741-1761 at elongation. As a result of the helix unwinding, the mid-thumb-ring linker loop extremity (residues 1750-1753, at pre-initiation), is displaced by 8 Å in the transition to elongation (**Fig. 6d**) and interacts with the priming loop residues 1414-1418 mainly through hydrophobic contacts (**Fig. 6d**)."

"Movements occurring between pre-initiation and elongation also differ between the two viral polymerases. For instance, the priming loop takes a completely different position and organization when it extrudes from the active site in the two cases (**Supplementary Fig.9**). Influenza polymerase priming loop, which is 35-amino acid long, has 17 residues extruded into the solvent in a disordered loop, and projects towards the PB1-PB2 interface helical bundle¹² (**Supplementary Fig.9b**). LACV-L priming loop, which is 20-amino acid long, orders itself at the surface of thumb ring and lid domains (**Supplementary Fig.9a**). Its extrusion is coupled with a *Peribunyaviridae* specific unwinding of the α -helix 72 of the mid domain into a loop, enabling an interaction between the priming loop and the extended mid/thumb ring linker (**Fig.6d**)".

10. *There are two polymerases in the asymmetric unit of the crystal structure. Are they identical? Please also compare them with the cryo-EM structure at the pre-initiation state.*

Comparison has been done and is now indicated in the text lines 439-445.

"The two polymerases of the crystallographic asymmetric unit are very similar with an RMSD of 0.79 Å over 2009 C α , although there is a slight difference in C-terminal domain orientation. They are also very similar to the cryo-EM structure at pre-initiation with RMSDs of respectively 3.022 Å and 2.951 Å over 2009 C α . A significant difference concerns the C-terminal linker and last α -helix that form a domain swapped dimer interaction in the X-ray structure but folds back on the ZBD in the monomeric cryo-EM structure."

11. *Page 16. "Before freezing and storing at -80°C, LACV-L bound to 3' (1-16) and 5' (9-16) vRNA were mixed with 5' (1-10) vRNA hook in a 1:2 molar ratio." Why not incubate them at the same time? Also, why choose 1:2 ratio? In addition, "The 5' (1-10) vRNA end was later soaked into crystals." Using the same ratio?*

We don't incubate the 5' (1-10) vRNA hook at the same time to avoid a mixed annealing of 3' (1-16) - 5' (9-16) and 3' (1-16) - 5' (1-10). This protocol was also used in Gerlach *et al.*, Cell, 2015. We add RNA in excess to favor complex formation. We have now added in the text the ratio of 5' (1-10) vRNA used for soaking (lines 415-416).

"The 5' (1-10) vRNA end was later soaked into crystals in 1:2 molar ratio."